# Dynamic spatiotemporal determinants modulate GPCR:G protein coupling selectivity and promiscuity

Manbir Sandhu [1,2] ✉, Aaron Cho[3], Ning Ma[1], Elizaveta Mukhaleva[1,4], Yoon Namkung [3], Sangbae Lee[1], Soumadwip Ghosh[1], John H. Lee[1], David E. Gloriam [5], Stéphane A. Laporte [3,6], M. Madan Babu [2,7] ✉ & Nagarajan Vaidehi [1,4] ✉

Recent studies have shown that G protein coupled receptors (GPCRs) show selective and promiscuous coupling to different Gα protein subfamilies and yet the mechanisms of the range of coupling preferences remain unclear. Here, we use Molecular Dynamics (MD) simulations on ten GPCR:G protein complexes and show that the location (spatial) and duration (temporal) of intermolecular contacts at the GPCR:Gα protein interface play a critical role in how GPCRs selectively interact with G proteins. We identify that some GPCR:G protein interface contacts are common across Gα subfamilies and others specific to Gα subfamilies. Using large scale data analysis techniques on the MD simulation snapshots we derive a *spatio-temporal code* for contacts that confer G protein selective coupling and validated these contacts using G protein activation BRET assays. Our results demonstrate that promiscuous GPCRs show persistent sampling of the common contacts more than G protein specific contacts. These findings suggest that GPCRs maintain contact with G proteins through a common central interface, while the selectivity comes from G protein specific contacts at the periphery of the interface.

G protein-coupled receptors (GPCRs) are membrane proteins that are critical in cell signaling. GPCRs play a pivotal role in intracellular communication and are highly tractable drug targets. GPCRs couple to the heterotrimeric G protein family to transduce an extracellular signal into various intracellular signaling cascades. The Gα subunit (Gα$_s$, Gα$_i$, Gα$_q$, Gα$_{12/13}$) of the trimeric G proteins is used to characterize the major signaling pathways activated by a G protein. The preferred signaling pathway for several GPCRs has been annotated in the IUPHAR "Guide to Pharmacology" database[1]. Several recent studies[2–12] highlight that GPCRs are generally promiscuous in their coupling to different G

protein subfamilies. Avet and colleagues have recently established that among 100 therapeutically relevant GPCRs, only 17 show selective signaling to one G protein subfamily[12]. For GPCRs that were formerly thought to signal through a single G protein subfamily, we now appreciate that different ligands can influence the same GPCR to signal through specific G proteins within and across G protein subfamilies[10]. Furthermore, certain pairs of GPCRs and G proteins that can biochemically coupled to each other can be limited by co-expression in specific cell types[13–17]. These factors collectively affect the inferred "selectivity" of a GPCR signaling response in vitro and in vivo.

[1]Department of Computational and Quantitative Medicine, Beckman Research Institute of the City of Hope, Duarte, CA 91010, USA. [2]Department of Structural Biology, Center for Data Driven Discovery, St. Jude Children's Research Hospital, Memphis, TN 38105, USA. [3]Department of Medicine, Research Institute of the McGill University Health Centre, Montreal, QC H4A 3J1, Canada. [4]Irell and Manella Graduate School of Biological Sciences, Beckman Research Institute of the City of Hope, Duarte, CA 91010, USA. [5]Department of Drug Design and Pharmacology, Faculty of Health and Medical Sciences, University of Copenhagen, 2100 Copenhagen, Denmark. [6]Department of Pharmacology and Therapeutics, McGill University, Montréal, QC H3G 1Y6, Canada. [7]MRC Laboratory of Molecular Biology, Francis Crick Avenue, Cambridge CB2 0QH, UK. ✉e-mail: msandhu@stjude.org; madan.babu@stjude.org; NVaidehi@coh.org

While extrinsic factors such as ligand identity and co-localization of a GPCR:G protein pair play a critical role in the signaling repertoire of a given GPCR, there are receptor-intrinsic structural factors that play a critical role in G protein coupling. Despite the explosion of high-resolution structures of GPCR and G protein co-complexes over the last decade[18–30], we are still decoding the mechanisms and structural determinants that define how specific GPCR:G protein complexes form. A G protein "barcode" for GPCR:G protein selective coupling has been identified[14] by analyzing the amino acid sequence of G protein residues that form contacts with GPCR residues in the interface[22,28,29]. At the same time, these studies have also highlighted the difficulty of deriving complementary determinants in the GPCR interface. Analysis of static structures and sequence information reflects that GPCRs have not evolved consensus sequences for recognizing G proteins[14,22]. More importantly, there is a paucity in the information on the structural dynamics mechanisms by which GPCRs promiscuously couple to multiple G proteins[14,31].

We postulate that the temporal persistence of the GPCR:G protein contacts are a paramount factor in G protein selectivity and promiscuity. The persistence of the GPCR:G protein residue contacts in the interface is influenced by the environment of residues in the neighborhood of these contacts. This persistence of the contacts is critical to determining the overall lifetime or the stability of the GPCR:G protein complex[32]. Given that the period of the vibrational modes of the GPCR:G protein non-bonded contacts (van der Waals and hydrogen bonds) is in the picosecond range, molecular dynamics (MD) simulations is a suitable method to probe the persistence of these contacts in lipid membrane bilayer and solvent environments.

In this study, we use atomistic MD simulations for 10 class A GPCR:G protein complexes that span both selective and promiscuous GPCRs and $G\alpha_s$, $G\alpha_i$, and $G\alpha_q$ protein subfamilies. We generate a map for the spatio-temporal persistence of GPCR:G$\alpha$ protein inter-molecular residue contacts and identify two critical types of GPCR:G$\alpha$ protein contacts: those that are persistent (sampled >20% frequency in the simulation) and (a) specific to each G$\alpha$ protein subfamily, which we name as "subfamily-specific contacts" and (b) common among complexes from all G$\alpha$ protein subfamilies ($G\alpha_s$, $G\alpha_i$, $G\alpha_q$), labeled as "common contacts". We applied data analysis techniques on 6 GPCR:G$\alpha$ protein contacts to generate a "spatio-temporal" code for G protein selectivity and validated the code on 3 GPCR:G protein complexes not used in the training. The GPCR:G$\alpha$ protein contacts that are important for selective coupling consist not only of the subfamily-specific but also some common contacts. The common contacts contribute to selectivity through differences in temporal persistence among GPCR:G protein residue contacts from different G protein subfamilies. All GPCRs sample the common contacts with higher persistence than the specific contacts, particularly the promiscuous triple mutant[33] β2-adrenergic receptor (ADRB2), which is the tenth GPCR:G protein compelx we have simulated. We validated our spatio-temporal code regulation by introducing switching mutations in the Gq-coupled muscarinic receptor M1 (CHRM1) to facilitate its coupling to the secondary G proteins, Gi and Gs using BRET-based assays.

## Results

### GPCRs exhibit a spectrum of coupling preference to G proteins
Recent work by Inoue et al.[11] and Avet et al.[12] have used biochemical and FRET-based experimental techniques to measure the pluri-dimensional coupling repertoire of GPCRs to multiple G$\alpha$ protein subtypes. The data generated by Inoue and colleagues was recently published in combination with existing data from the IUPHAR "Guide to Pharmacology" database on the GPCRdb website[34,35]. This data summarizes G protein coupling for the available GPCRs into categorical rankings for "no coupling", "secondary coupling", and "primary coupling." We have similarly categorized (see "Methods") the measured coupling data provided in "Table S1D: double normalized $E_{max}$

values" from the Avet et al. study to consolidate GPCR:G protein coupling information across these three resources[36].

The experimental methods used by Avet et al. directly measure activation of G proteins, whereas the signal measured in the Inoue study undergoes amplification between the initial receptor:transducer coupling event to the measured biochemical outcome. Moreover, the Inoue et al. study used a chimeric Gq protein in which the last six C-terminal amino acids were substituted for different G protein sequences to test coupling of GPCRs to these chimeric Gq proteins, rather than the wildtype G proteins. For this reason, we assigned different weights to the categories describing binding from these two studies and the "Guide to Pharmacology" resource (see "Methods"). Among 404 non-olfactory class A GPCRs, G protein coupling information was available for 267 receptors across these three datasets. The number of distinct G protein subtypes measured also differed between datasets and in the "Guide to Pharmacology" resource, G protein coupling was only provided as categorical descriptions for the four major G protein subfamilies. Therefore, we generated a composite score describing the G protein coupling repertoire for these 267 receptors for each of the 4 G protein subfamilies (see "Methods"). We combined these scores across the four subfamilies to generate an average "promiscuity" index (Supplementary Data 1). The results of this promiscuity index are summarized in the heatmap in Fig. 1 to show both the strength of the evidence for coupling and the promiscuity of coupling for 267 GPCRs.

Our analysis shows GPCRs exhibit a spectrum of coupling behavior towards G proteins ranging from selectivity to a single G$\alpha$ protein subfamily (e.g. ADGRG2 ($G_q$), FZD10 ($G_{12/13}$), and P2RY12 ($G_i$) receptors) to promiscuous coupling to multiple G$\alpha$ protein subfamily members (e.g. LPAR1 ($G_i$, $G_q$, $G_{12/13}$), LPAR2 ($G_s$, $G_i$, $G_q$), BDKRB2 ($G_s$, $G_i$, $G_q$, $G_{12/13}$), GPPR4 ($G_s$, $G_i$, $G_q$, $G_{12/13}$), GPR68 ($G_s$, $G_i$, $G_q$, $G_{12/13}$) receptors).

Here we study the role of structural and dynamic factors that contribute to the selective and promiscuous coupling of G proteins by GPCRs. The need for understanding dynamic factors of selectivity become evident when comparing the three-dimensional structures of different GPCRs coupled to the same G protein subfamily. Comparison of the orientation of the C-terminal α5 helix of $G\alpha_s$, $G\alpha_i$, and $G\alpha_q$ proteins in their respective GPCR:G protein complexes show an ensemble of different orientations for the α5 helix as it inserts into the GPCR intracellular cavity (Supplementary Fig. 1a–c). Even for a single GPCR such as $NTS_1R$, multiple binding conformations for the same G$\alpha$ protein have been observed within cryo-EM structures[37,38] and may hint at a conformation ensemble modulating the signaling output through the Gi protein (Supplementary Fig. 1d). This highlights the importance of considering the dynamics of intermolecular contacts in the GPCR:G protein interface to understand the mechanisms of recognition of G proteins by GPCRs.

### Structural dynamics reveals GPCR:G protein contacts that are specific and common across G protein subfamilies
We performed molecular dynamics (MD) simulations of GPCR:G protein complexes in lipid bilayer to understand how the persistence of the intermolecular contacts between GPCR and G protein residues contribute to G protein coupling selectivity and promiscuity. The computational workflow is shown in Fig. 2. We combined MD simulation trajectories and data analysis techniques to map the location and duration of the GPCR:G$\alpha$ protein intermolecular contacts. We performed 800 ns–1 μs of MD simulations on each of the five replicates for each of the 6 GPCR:G protein complexes used for training the model (complex, PDB IDs: ADRB2:Gs, 3SN6; ADORA2A:miniGs, 6GDG; ADORA1:Gi2, 6D9H; HTR1B:Go1, 6G79; CHRM1:G14, 6OIJ; HTR2A:miniGq, 6WHA), starting from their respective X-ray or electron microscopy resolved structures (Fig. 2a). We selected the last 200 ns window of each simulation replicate to combine into an ensemble trajectory

 

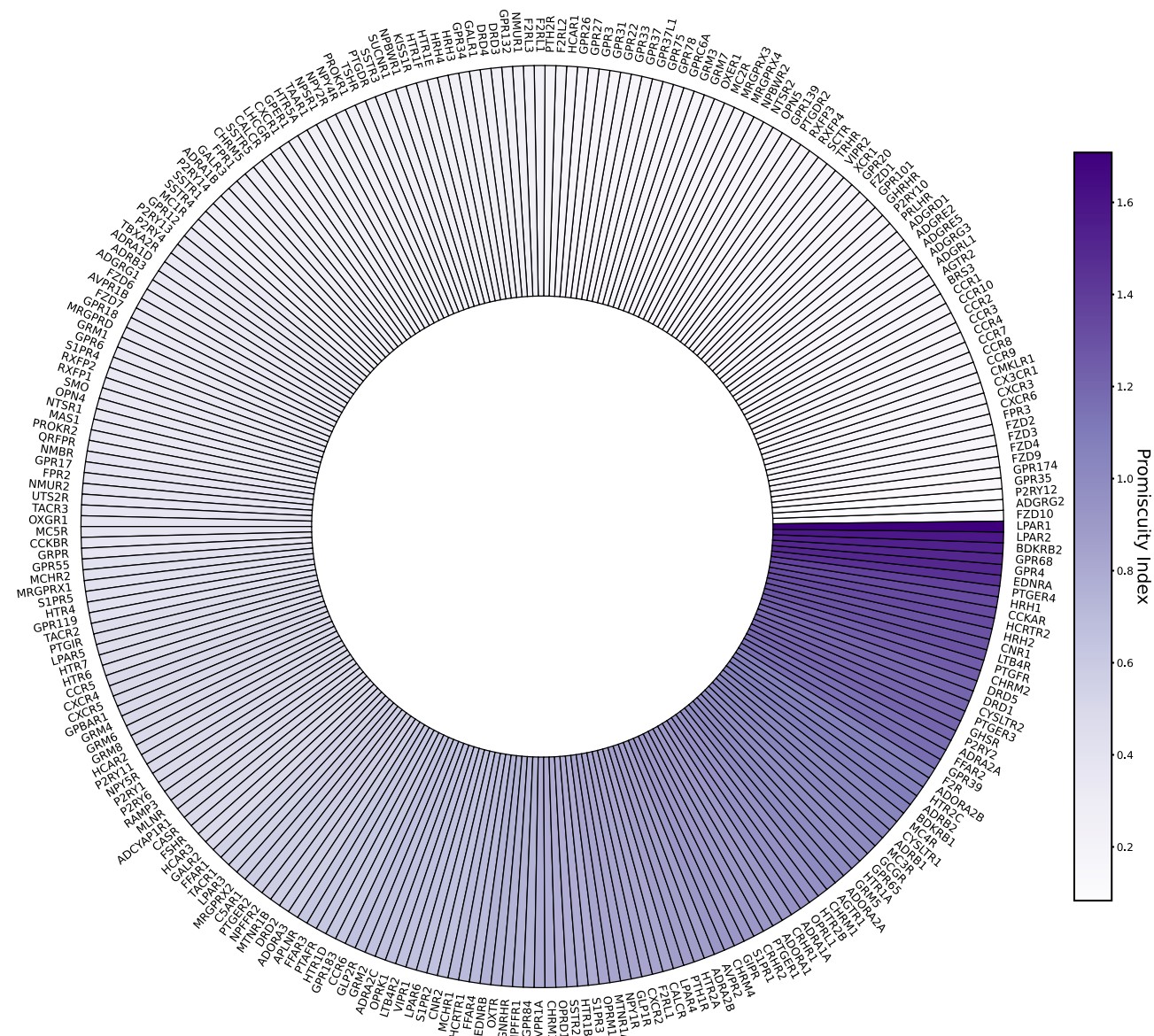

**Fig. 1 | GPCRs exhibit a spectrum of coupling strength to G-proteins; from selective interaction with a single G protein to promiscuous coupling across G protein subfamilies.** GPCRs studied in Inoue et al.[11] and Avet et al.[12] along with those annotated in the IUPHAR database were compared to assess GPCR coupling selectivity across major G protein subfamilies. We calculated a "promiscuity index" and ranked GPCRs based on this index, taking into consideration the strength of evidence for coupling based on the type of assay or reporting used. The circularized bar plot displays the spectrum of promiscuity as denoted by the color heatmap at the right of the figure. Source data are provided as Supplementary Data 1.

totaling 1 μs of simulation time. Within this 1 μs ensemble, we calculated all sidechain-to-sidechain contacts between the GPCR and Gα subunit of the G protein using the "getcontacts" python script[39] (Fig. 2b). Each pairwise residue contact from GPCR to G protein was labeled based on GPCRdb numbering scheme[40] and Common G protein Numbering[41]. We focused on non-covalent contacts between sidechain atoms to analyze the effect of specific protein sequences. This framework allows for interpreting the residue contact pairs from a spatial, temporal, and chemical perspective.

We did not consider residue contacts that were formed by regions of the GPCR that were not represented in every complex (Gs, Gi, Gq – see "Methods"). This resulted in 764 total pairwise contacts for further analysis (Supplementary Data 2, Fig. 2c). Out of the 764 contacts we analyzed the contacts that showed a persistence of >20% of the simulation time, which represents a duration of at least 40–200 nanoseconds. The persistent contacts that were observed only in interactions with a specific subfamily of Gα proteins are termed as

"subfamily specific contacts." We identified 24 such contacts for Gs, 13 contacts for Gi, and 18 contacts for Gq (Supplementary Fig. 2a–c). The persistent residue contacts observed in all Gα protein subfamilies are termed as "common contacts" and we identified 23 such contacts (Fig. 3c and further below).

We identified distinct trends when evaluating which secondary structural elements (SSE) of GPCRs and G proteins contribute to the common and subfamily specific contacts. Each pie chart denotes the percentage of contacts coming from different SSEs of the GPCRs and G proteins for subfamily specific (Supplementary Data 3, Fig. 3a, b) and common contacts (Fig. 3d). Gs-coupling GPCRs involve fewer SSEs to interface with G proteins than Gi-coupling and Gq-coupling GPCRs. In contrast, the Gs proteins interface with their GPCR partners through a much larger and diverse set of SSEs compared to Gαi and Gαq proteins in complex with their GPCR partners.

The GPCR residues contributing to the Gs subfamily specific contacts (in ADRB2 and ADORA2A) are often found in transmembrane

# a

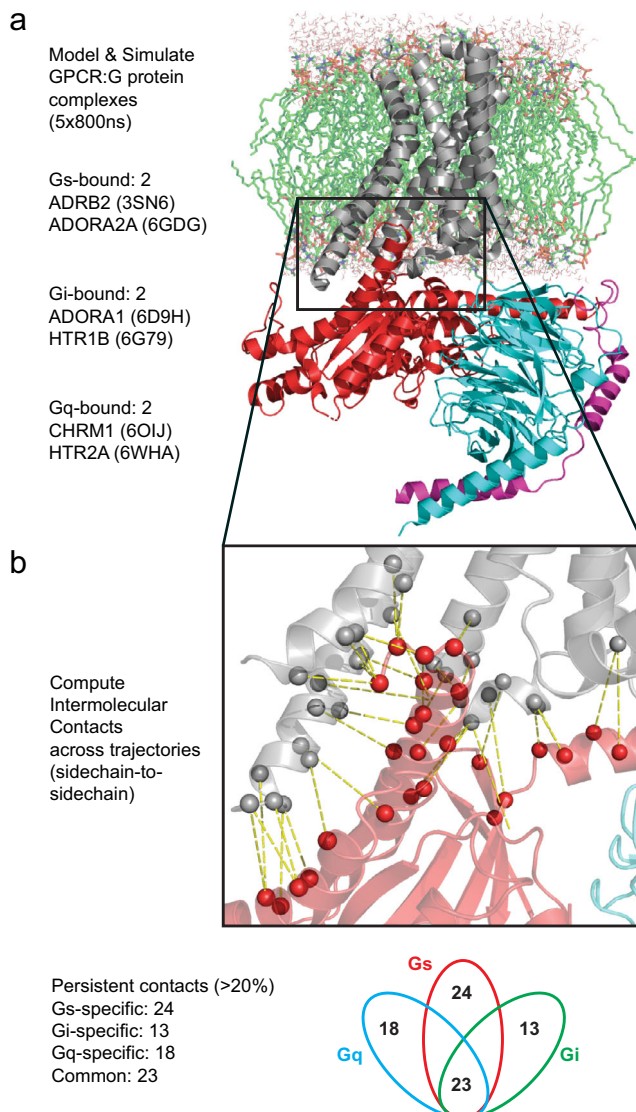

Model & Simulate GPCR:G protein complexes (5x800ns)

Gs-bound: 2
ADRB2 (3SN6)
ADORA2A (6GDG)

Gi-bound: 2
ADORA1 (6D9H)
HTR1B (6G79)

Gq-bound: 2
CHRM1 (6OIJ)
HTR2A (6WHA)

# b

Compute Intermolecular Contacts across trajectories (sidechain-to-sidechain)

Persistent contacts (>20%)
Gs-specific: 24
Gi-specific: 13
Gq-specific: 18
Common: 23

**Fig. 2 | Workflow for extracting the spatiotemporal heat map for the residue contacts in GPCR:G protein interface. a** We used six crystal and cryo-EM structures of GPCR:G protein complexes to model the interactions of two Gs-coupled, two Gi-coupled, and two Gq-coupled GPCR:G protein pairs. Atomistic MD simulations were performed as five replicates, each replicate extended to >800 ns of simulation time for each of these GPCR:G protein complexes. The last 200 ns window of each replicate was combined into a 1 µs ensemble trajectory used for further analysis. Source data are hosted at GPCRmd.org. **b** The sidechain-to-sidechain intermolecular contacts between GPCR and Gα proteins were computed for each of the six complexes using the get_contacts.py method. We obtained 764 contacts for analysis and identified "persistent contacts" as those sampled for >20% frequency in one or both systems representing each class (Gs, Gi, Gq). Among these persistent contacts, we identified a subset that appear uniquely in interactions of one Gα protein family ("specific contacts") and those which are found across all of the Gα protein subfamilies (Gs, Gi, and Gq; "common contacts").

helix 5 (TM5) (58.3%), followed by ICL2 (25%), then similarly from TM3 and Helix 8 (H8) (8.3% each). The Gα$_s$ residues involved in subfamily specific contacts arise predominantly from the C-terminal α5 helix (H5) (50%) and the remainder from several small loops, helices, and beta-sheets within the G protein alpha-helical domain. Gi interactions are marked by a large GPCR interface with roughly similar contributions from several SSEs: ICL1, TM2, TM5, TM6 (7.7% each); ICL2, TM7, H8 (15.4% each); TM3 (23.1%). Like Gα$_s$, most subfamily specific contacts formed by Gα$_i$ arise from H5 (76.9%), with a few interactions from

nearby loops h3s5 and s2s3 and beta-sheet S3 (7.7% each). The GPCR residues involved in Gq subfamily specific contacts are primarily located in TM2 (33.3%) and ICL2 (27.8%), with a few residues from TM3 (5.6%), TM4, TM6, and H8 (11.1% each). The Gα$_q$ protein also contributes to a large percentage of contacts using H5 (72.2%), with few interactions arising from key loops, hns1 (16.7%) and s2s3 (11.1%).

There are 23 common contacts sampled with different temporal frequencies observed across G protein subfamilies (Fig. 3c). Sequence alignment of the GPCR and G protein residues found a high degree of conservation of residues forming the common contacts at these positions across the 6 GPCRs and 5G proteins used in this study (Supplementary Fig. 2d–e). Most residues contributing to common contacts are found in ICL2 (43.5%), TM3 (21.7%), and TM6 (17.4%) of GPCRs and H5 of G proteins (87%) (Fig. 3d). It should be noted that although these common contacts would have been discounted as not contributing to G protein selectivity based on sequence analysis, we show in the next section that they indeed do so by modulating their temporal frequencies of these contacts.

## Deriving the spatiotemporal code for G protein selectivity by GPCRs

To identify the GPCR:G protein residue contacts that confer selectivity to G protein coupling, we performed the data analysis procedure shown in Fig. 4a. The 764 intermolecular contacts from each trajectory were accumulated into a binary dataset by one-hot encoding the various contacts as features (columns) of the dataset, and the rows being each frame of the trajectory. Within each row, if a particular contact is present within that frame, the feature is noted by a "1", and if absent it is marked as "0". Thus, this binary dataset delivers a "fingerprint" of contacts for each frame of the trajectory (Supplementary Data 2).

Each row of the dataset is assigned a class value of "Gs" (for frames of the ADRB2:Gs and ADORA2A:miniGs simulations), "Gi" (for frames of the ADORA1:Gi2 and HTR1B:Go1 simulations), or "Gq" (for frames of the CHRM1:G11 and HTR2A:miniGq simulations), depending on which simulation the row represents. By segregating rows into the appropriate classes, we performed a linear discriminant analysis (LDA) to identify a singular value decomposition of the contact space that segregates each class from one another (Fig. 4a). We trained the LDA classifier using the entirety of the contact fingerprint dataset from the six simulations (Fig. 4b) and have shared the resulting outputs from the model (Supplementary Data 4–8).

We used the LDA classifier as a feature ranking method to determine which contacts contribute most prominently for the distinct interaction signature of different GPCR:Gα protein complexes. We used the composite weight computed in Supplementary Data 5 ("wGx" values) to identify the top 10 contacts for each class of G protein interactions which we have denoted as the "Spatiotemporal code" for selectivity. These G protein selectivity contacts are provided in a matrix (Fig. 4c, Supplementary Data 6) with GPCR residues on the vertical axis and G protein residues on the horizontal axis. For each of these contacts, we have also enumerated which contacts were observed in the three-dimensional structures used for starting the MD simulations, and which, if any, residues are missing from the original PDB files (Supplementary Data 6). Each square is colored to represent the G protein class (Gs in red, Gi in green, Gq in blue). 13 of 30 selectivity code contacts are from the set of "common contacts" identified earlier, with 5 in the Gs spatiotemporal code, 4 in Gi, and 4 in Gq. We observe that 12 out of 30 contacts in the spatiotemporal code belong to the subfamily specific contacts with two contacts identified in Gs, 4 in Gi, and 6 in Gq. The remaining contacts found in the code (5 of 30) are sampled highly in more than one subfamily, but not across all three G protein subfamilies. We do observe one distinct residue contact pair that has been identified as both Gi and Gq selective, "3 × 53" and "G.H5.19" (cyan square, Fig. 4b). This contact is ranked higher in the Gi class and is sampled at a higher frequency in the Gi-coupled complexes

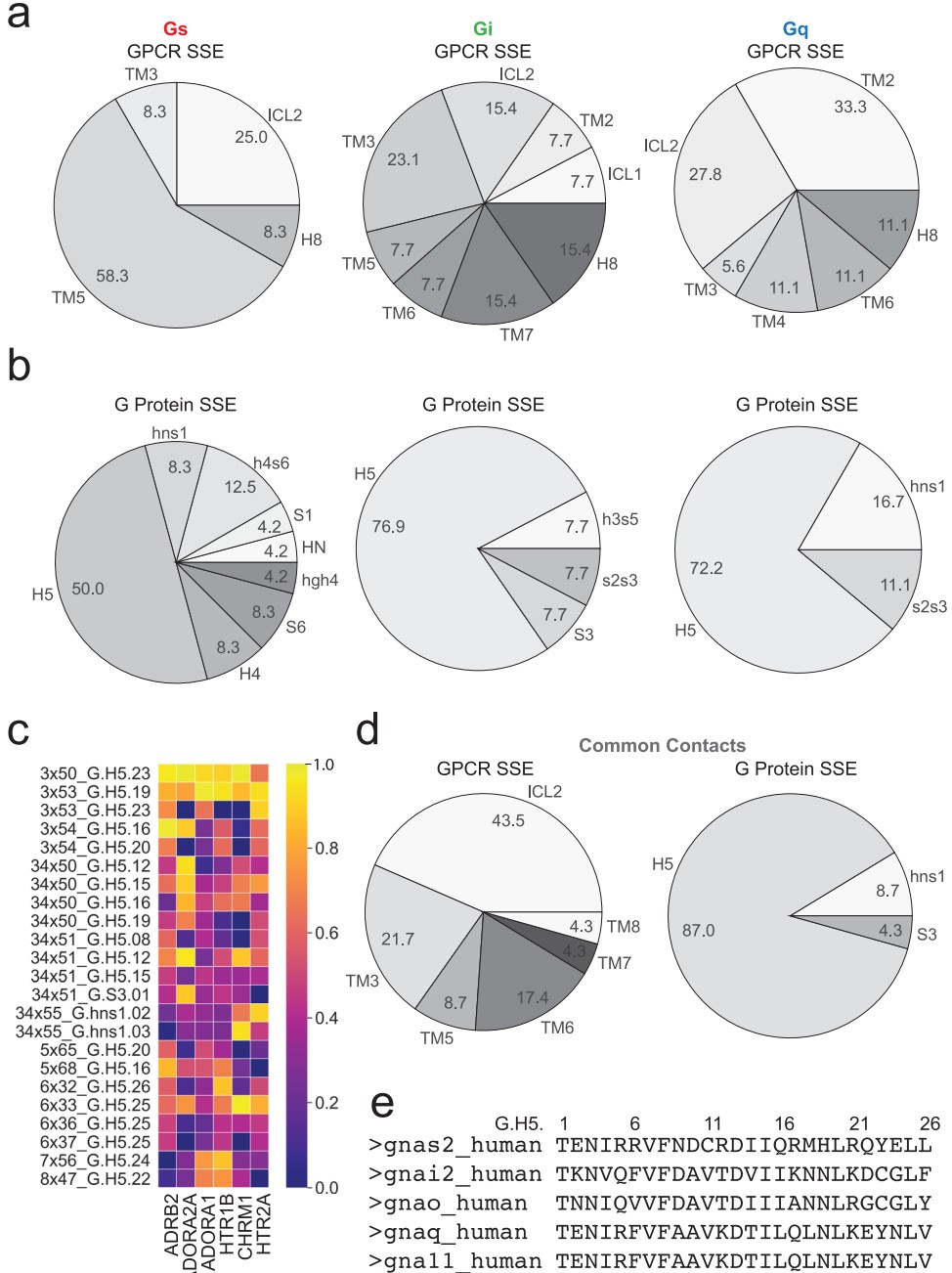

**Fig. 3 | Unique signatures of GPCR and G protein structural regions involved in contacts from different G protein families.** The location of the G protein family "specific contacts" mapped to the various GPCR and G protein secondary structural elements (SSE). Source data for all plots are provided as Supplementary Data 2. **a** The percentage of GPCR:G protein contacts specific to the Gs (left), Gi (center), and Gq (right) interaction arising from each SSE of the GPCR is shown. **b** Similar pie chart displaying the percentage of contacts from specific SSEs of the G proteins. The residue numbering system used is from the "common G protein numbering" scheme developed by Flock et al.[41]. **c** The temporal frequency of each GPCR:G protein "common contact" is displayed, indicating the frequency of each contact for each GPCR system. **d** The amino acids found among "common contacts" are shown in the pie charts, labeled by the SSE in which they are found. **e** FASTA sequence alignment of G protein C-terminal H5 helix for G proteins used in MD simulations.

than the Gq-coupled complexes showing that temporal frequency of the GPCR:G protein contacts may also play a critical role in selectivity.

For the Gs-selective LDA contacts, 5 of the 10 contacts are formed within the original PDBs of either ADRB2 (PDB 3SN6) or ADORA2A (PDB 6GDG). Those contacts are sampled at frequencies of 23.9-99.6% of the simulation duration, while the contacts newly formed within the simulation are sampled at frequencies of 9.8-93.4% of the simulation. For Gi-selective LDA contacts, 4 of the 10 are found within one or both original PDB structures for ADORA1 (PDB 6D9H) or HTR1B (6G79). Those contacts are sampled at frequencies of 29.5-65.4% of the simulation, while newly formed contacts are sampled at 17.1-96.1% of simulation duration. Among these Gi selective contacts, the residue 5×71 was missing from the HTR1B PDB file. For Gq-selective LDA contacts, 2 of the 10 contacts were found within the PDB structures of CHRM1 (PDB 6OIJ) and HTR2A (PDB 6WHA). Those contacts are sampled at frequencies of 70.0% and 89.1% of the simulations, and the newly formed contacts are sampled at frequencies ranging from 25.9% to 89.2% of the simulation.

To identify the structural location of selective contacts in the spatiotemporal code matrix, we mapped the GPCR residues (colored

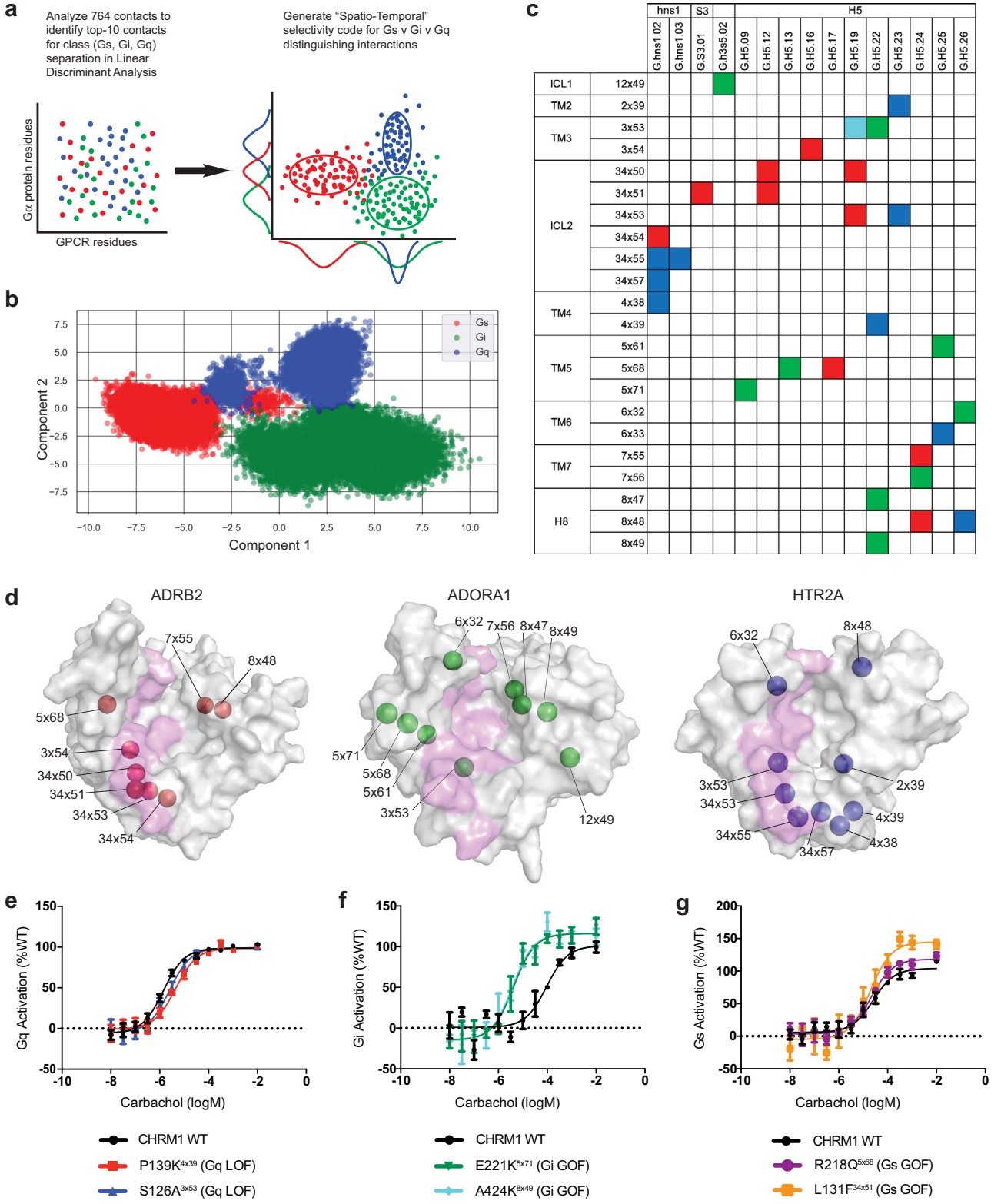

and labeled spheres) from contacts found in the spatiotemporal code onto the structure of a representative GPCR from the set of simulated receptors (Fig. 4d). We find that GPCR residues involved in selective G protein binding are found predominantly at the periphery of the intracellular surface while those contacts common to all G protein subfamilies (light-pink colored surface) are found at the central core of the GPCR surface, spanning a "vertical" interface from TM6 to ICL2. This representation shows clearly that GPCR residues involved in "common contacts" are in a highly conserved interface across GPCRs

with varied G protein coupling preferences and the sequence identity of these positions is highly conserved among the GPCRs used in this study (Supplementary Fig. 2d).

**Experimental validation of residue positions in the spatio-temporal code that modulate selectivity to cognate and non-cognate G proteins in CHRM1**

We tested the importance of the residue positions identified in the spatio-temporal code in their ability to modulate selective G protein

**Fig. 4 | Identifying G protein selectivity determinants using linear discriminant analysis of spatiotemporally resolved GPCR:G protein contacts. a** GPCR:Gα protein contacts from each frame of the MD simulations were one-hot encoded into a binarized interaction fingerprint and used to train a linear discriminant classifier and identify features (intermolecular contact pairs) that distinguish Gs, Gi, and Gq interactions. **b** The projection of each frame of the MD simulations, colored by G protein interaction (red, Gs; green, Gi; blue, Gq) are shown projected into the two-dimensional deconvoluted space (Component 1, Component 2) determined by the linear discriminant analysis. Source data are provided as Supplementary Data 4. **c** The top ten pairwise contacts which contribute highly to the interaction signature of each G protein family are displayed in the spatio-temporal barcode as the GPCR (rows) and G protein (columns) residues that are found in the distinguishing pairwise contacts for each G protein interaction (red, Gs; green, Gi; blue, Gq). One contact ('3×53:G.H5.19') is shared among Gi and Gq type interactions, and is colored in "cyan." **d** GPCR residues involved in G protein selectivity are displayed on the

intracellular surface of a Gs (ADRB2, red spheres, left), Gi (ADORA1, green spheres, center), and Gq-coupled (HTR2A, blue spheres, right) GPCR. Residues that are found in "common contacts" across G protein subfamilies are shown as a magenta-colored surface. **e** BRET-based Gq-activation sensor was used to measure CHRM1 (WT, P139K and S126A mutants) activation of Gq heterotrimers in the presence of carbachol ($n = 3$ biologically independent samples, data are presented as mean ± SD for each concentration). **f** BRET-based Gi-activation sensor was used to measure CHRM1 (WT, E221K, and A424K mutants) activation of Gi heterotrimers in the presence of carbachol (n = 3 biologically independent samples, data are presented as mean ± SD for each concentration). **g** BRET-based EPAC sensor was used to measure CHRM1 (WT, P139K and S126A mutants) activation of Gs signaling and cAMP accumulation in the presence of carbachol and 500 nM of Gq-protein inhibitor YM-254890 (n = 3 biologically independent samples, data are presented as mean ± SD for each concentration).

coupling. We used the muscarinic acetylcholine receptor M1 (CHRM1) as our test case to determine if modifying the amino acids at selectivity positions identified for Gs, Gi, and Gq interactions could change the behavior of CHRM1-mediated G protein activation. We prioritized contacts from the spatiotemporal code that were identified from G protein family-specific contacts, where amino acid identity was not conserved in receptors with coupling preference to other G proteins. We mutated residues in the CHRM1 to amino acids found at those same positions in either the ADRB2, HTR1B, or ADORA1 receptors.

To disrupt interaction with the cognate Gq protein, we mutated Gq-selective residue positions 4×39 and 3×53 in CHRM1 to their amino acid identities in ADRB2; P139K$^{4×39}$ and S126A$^{3×53}$. Both of these mutants show a significant right-shift in the EC$_{50}$ curve for carbachol-mediated CHRM1 activation of Gq (Fig. 4e), demonstrating a loss in potency for Gq activation (logM of −5.851 to −5.351; $p$-value = 0.0011**, and to −5.595; $p$-value = 0.0117*, respectively). We enhanced the activity of the CHRM1 to activate Gi protein by introducing swapping mutations at the predicted Gi-selective positions 5×71 and 8×49 with the corresponding amino acids from HTR1B and ADORA1, respectively. Both the mutants E221K$^{5×71}$ and A424K$^{8×49}$ improved potency of Gi activation by shifting the EC$_{50}$ of carbachol-mediated CHRM1 activation of Gi from logM of −4.025 to −5.374 ($p$-value = 0.0007***) and to −5.428 ($p$-value = 0.0004***), respectively (Fig. 4f). These mutants also significantly increased the Emax of CHRM1 activation of Gi by 30% ($p$-value = 0.0132*) and 31% ($p$-value = 0.0113*) compared to WT CHRM1. Finally, we modified Gs-selective residue positions R218$^{5×68}$ and L131$^{34×51}$ to the corresponding amino acids in ADRB2 to enhance coupling of CHRM1 with Gs protein. Because CHRM1 inefficiently engaged the Gs BRET sensor[12], we used the downstream EPAC-BRET sensor[42] to measure cAMP production by CHRM1 and mutants. We also included an inhibitor of Gq protein, YM-254890, to ensure that Gs-dependent cAMP accumulation was independent on Gq and Ca$^{2+}$ cross-talk. Both mutants demonstrated a significant increase in the Emax of Gs-dependent cAMP production following carbachol treatment (17% ($p$-value = 0.0242*) for R218Q and 51% ($p$-value = 0.0022**) for L131F; Fig. 4g). None of these substitutions significantly altered mutants' expression as compared to WT CHRM1 (Supplementary Fig. 3a).

**Residue positions in the spatio-temporal code show lower propensity for natural variation**

To validate the functional importance of these LDA-derived spatio-temporal code residue positions, we calculated the mean variation of all residue positions identified as missense variants in the gnomAD v3.1 population database[43] for the six GPCRs studied here. We compared the average number of variants per residue position identified among the gnomAD population at positions that are part of the LDA-derived spatiotemporal code for the given receptor, and those

positions not found in the code. We observe that GPCR residues within the LDA spatiotemporal code show lower variation in the population, suggesting that these positions are likely under selective pressure (Supplementary Fig. 3b, Supplementary Data 8). This points to the functional role of the spatiotemporal code residue positions that are in the GPCR:G protein interface compared to other interface residues.

We further validated the importance of the LDA-derived contacts by analyzing the measured effect of mutations on the GPCR residue positions generated using deep mutational scanning measurements by Jones et al.[44] that examined the role of every amino acid substitution on ADRB2. The authors use a barcoded (DNA sequence) transcriptional reporter that measures Gs-signaling output through a multimeric cAMP response element that transcribes the unique barcode. The authors mutated each amino acid position of the ADRB2 to each of the 19 alternate amino acids and measured signaling activity via the transcriptional reporter at basal (non-stimulated), 150 nM (EC$_{50}$ dose), 625 nM (EC$_{100}$ dose), and 5 µm (saturating dose) concentrations of isoproterenol. We examined the normalized activity for all mutants globally to be "1.75" activity units which suggest an overall high level of mutational tolerance at most positions and substitutions. We observe that the ADRB2 residue positions found in the spatiotemporal code for Gs-selectivity are well below the global average mutant activity. Most positions (34 × 50, 34 × 53, 34 × 54, 5 × 68, 7 × 55) were measured below 1.0 activity units (Supplementary Fig. 3c). These results demonstrate that ADRB2 residue positions found among Gs-selectivity conferring contacts are intolerant to mutation, and can strongly affect Gs signaling.

**Test set of GPCR:G protein complexes further validate the spatio-temporal code**

Lastly, we determined how well the LDA model can identify contacts for a test set of simulations performed for the Gs-coupled Dopamine D1 Receptor (D1DR), Gi-coupled Cannabinoid 1 Receptor (CB1R), and Gq-coupled Histamine H1 Receptor (HRH1) (Supplementary Figs. 4d–f). These receptor complexes were not included in the six GPCR:G protein complexes used for the derivation of the spatio-temporal code. We observe that simulation snapshots project onto similar space in the LDA Component 1 and Component 2 vectors, with the Gs-coupled DRD1 aligning best with the Gs class of contacts (Supplementary Fig. 3d). The CB1R and HRH1 simulations aligned prominently with their Gi-coupled and Gq-coupled contacts, respectively, but we also observe overlap into other G protein classes (Supplementary Fig. 3e–f). This overlap may correspond to the more promiscuous nature of CB1R and HRH1 receptors, as demonstrated in the promiscuity index (Fig. 1). In addition, predictive models based on methods such as LDA can be refined as more structures of GPCRs with full length, non-chimeric G proteins become available in the future.

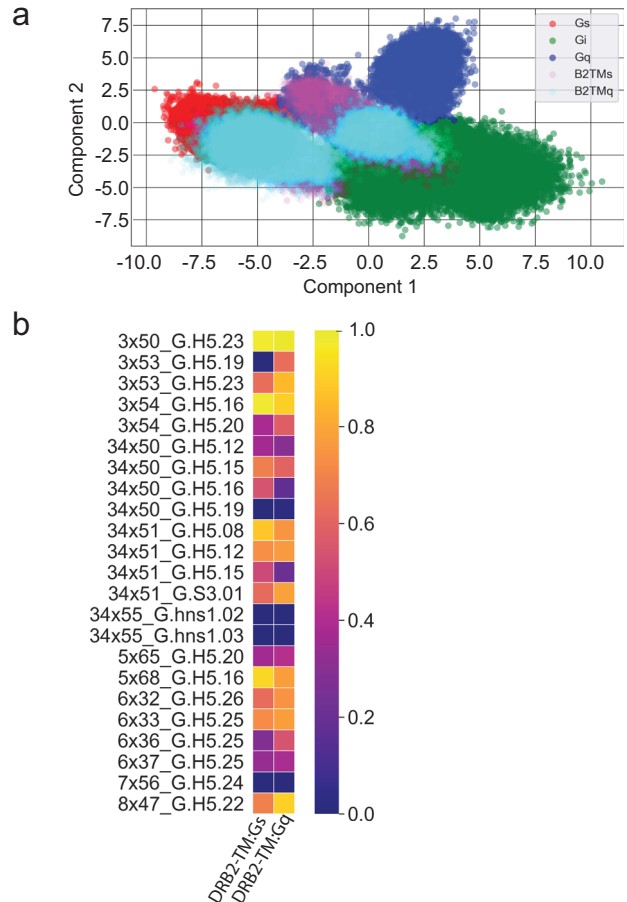

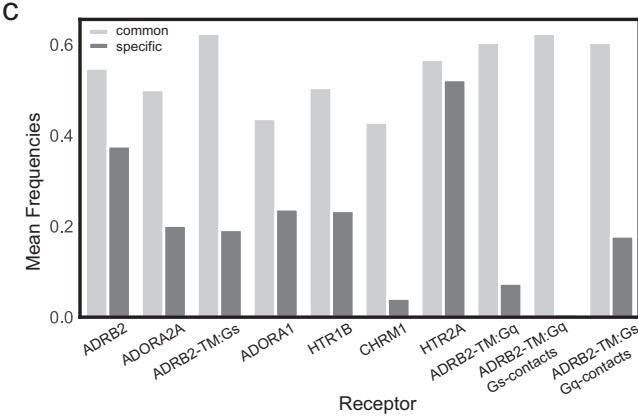

**Fig. 5 | The GPCR:G protein contact landscape of promiscuous GPCRs. a** Analysis of promiscuous GPCRs in the framework of the LDA model trained on data from the original six structurally resolved GPCR:G protein complexes. The promiscuous GPCR:G protein interactions are plotted using the feature weightings used to compute Component 1 and Component 2 from the LDA analysis performed in Fig. 4. The promiscuous GPCRs are distinguished by color and denoted in the legend: ADRB2-TM with Gs ("B2TMs", magenta), ADRB2-TM with Gq ("B2TMq", cyan). The plotting of these promiscuous receptor interactions lie predominantly in the common space of the LDA model, suggesting the promiscuous receptor interactions with G proteins are represented by mostly non-distinguishing contacts. **b** Heatmap of contact frequencies for "common contacts" sampled in the promiscuous ADRB2-TM receptor. Source data are provided as Supplementary Data 3. **c** The promiscuous GPCRs studied here show stronger sampling (measured by mean frequency per residue) within the subspace of "common contacts" pairs (light gray), and less frequent sampling of G protein family "specific contact" pairs sampled within the G protein family to which they are coupled (medium gray). For the promiscuous ADRB2-TM, we calculated the mean frequency of subfamily specific contacts for the G protein-coupled within the simulation (ADRB2-TM:Gq, ADRB2-TM:Gs), and the mean frequency of subfamily specific contacts for the opposite G protein family, while coupled to one of the G proteins (ADRB2-TM:Gq with Gs-contacts, ADRB2-TM:Gs with Gq-contacts).

TM:Gs fingerprints into the correct class ("Gs") and observe an accuracy of 57.41%. The ADRB2-TM:Gs fingerprints when tested for sorting into class of "Gq", the accuracy of the classifier was 30.92%. These results suggest that the promiscuous ADRB2-TM may sample conformation features that represent both "Gs" and "Gq" types of interactions and is able to accommodate the Gs protein in both types of interfaces. We also tested how well the ADRB2-TM:Gq complex represents a "Gq" specific signature and projected the fingerprints of this simulation onto the LDA space (Fig. 5a, cyan circles). We similarly observe that these fingerprints do not segregate with the "Gq" signature, and when we tested the LDA classifier, we observed only a 3.77% accuracy assigning these fingerprints to the "Gq" class. Surprisingly, when testing the classifier to sort into the "Gs" class, the accuracy rose to 84.94%. From evaluating the contact frequencies of these ADRB2-TM simulations we identified a strong preference for each complex to sample the subset of common contacts (Fig. 5b).

To understand further why the promiscuous ADRB2-TM simulations poorly segregate with the "Gs" and "Gq" signatures we traditionally expect, we analyzed how each GPCR samples common and subfamily specific contacts in each simulation. We computed the mean frequency per residue for residues from the subset of "common contacts" and "G protein subfamily specific" contacts for each of the eight GPCR:G protein simulations shown in Fig. 5c (six original simulations from the LDA model and two simulations of the promiscuous ADRB2-TM with Gs and Gq proteins). We also calculated the contribution of the common and specific contacts to the GPCR:G protein non-bonded interaction energies averaged over the MD trajectories (Supplementary Fig. 3g). Surprisingly, we observe that all the GPCR:G protein complexes studied show on average a higher frequency or duration of contact (~45–65%) within the common subset as compared to the G protein-specific subset (~5–55%) (Fig. 5c). Most of these receptors also show a similar trend with the calculated average interaction energies for these contacts (Supplementary Fig. 3g). We observe that the ratio of the mean frequency for common versus specific contacts is most prominent for the two promiscuous receptor simulations, ADRB2-TM:Gs and ADRB2-TM:Gq. Because both ADRB2-TM simulations showed higher than expected sorting into their alternative classes with the LDA classifier, we also evaluated the mean frequency of sampling for the alternative G protein subfamily specific contacts (ADRB2-TM:Gs sampling of Gq-specific contacts, and ADRB2-TM:Gq sampling of Gs-specific contacts). We observe that ADRB2-TM:Gs samples very little of the Gq-specific contact space, while the ADRB2-TM:Gq complex indeed is able to sample more Gs-specific contacts than Gq-specifc contacts, but still at a much lower mean frequency than common

## Promiscuous GPCRs sample more of the common contacts than G protein subfamily-specific contacts

In our previous study of the structural dynamics of promiscuous receptors, we showed that GPCRs have latent intracellular cavities that can be engineered to reshape and signal through multiple G proteins[33]. We used this engineered Q142K$^{34×54}$-R228I$^{5×67}$-Q229W$^{5×68}$ triple-mutant of ADRB2, denoted as ADRB2-TM, which is promiscuous to Gs and Gq coupling with relatively similar potency[33], to perform MD simulations with both the Gs and Gq protein heterotrimers. We analyzed how these contacts compared to the signature identified from the less-promiscuous GPCRs used in our LDA classifier. We projected the fingerprints from the ADRB2-TM:Gs complex onto our LDA space (Fig. 5a, magenta circles) and observe that these fingerprints do not segregate completely with the "Gs" signature (red circles). In fact, there is sufficient sampling of the 2D space representing the "Gq" and "Gi" signatures. We tested the ability of our LDA classifier to sort the ADRB2-

contacts. We expanded this analysis to compare the sampling of common contacts compared to any other contacts sampled within each class (Supplementary Fig. 3h). From this analysis, we still observe a much higher mean frequency of interaction through the common contacts (~45–65%) as compared to other contacts (~30–50%) sampled for each GPCR:G protein class. Interestingly, as seen in Supplementary Fig. 3b, the interaction energies coming from both common and specific contacts are similar in magnitude in the eight receptors studied here except in HT2A and in the promiscuous receptor ADRB2-TM. This suggests that promiscuous GPCRs sample more of the common contacts and likely derive their coupling strength from them more than the specific contacts.

## Discussion

We put these results forward as a model for selective and promiscuous GPCR:G protein coupling; promiscuous GPCRs can readily interface multiple G proteins using residues at the core of the GPCR intracellular surface. Selective GPCRs likely engage cognate G proteins through the common contacts and further stabilize the complex by forming selectivity conferring contacts located at the periphery of the GPCR:G protein interface.

Our analysis of experimental G protein coupling data[11] showed that GPCRs exhibit a continuous spectrum of coupling strengths to several G proteins, ranging from promiscuity across all four subfamilies ($G\alpha_s$, $G\alpha_i$, $G\alpha_q$, and $G\alpha_{12/13}$) to highly selective coupling of single G protein subtypes. We find that there is no preference for GPCRs within a given receptor family to either be G protein selective or promiscuous. This suggests that the GPCR intracellular surface maintains a level of plasticity to evolve in either direction, i.e., towards selectivity or promiscuity.

Analysis of the detailed spatiotemporal resolution of contacts derived from the dynamics of 6 GPCR:G protein complexes showed two important types of persistent contacts: (i) those that are G protein subfamily-specific and (ii) contacts that are common across G protein subfamilies. Linear Discriminant Analysis of all sampled contacts yielded the "spatiotemporal code" for G protein selectivity. The contacts that contribute to G protein selectivity are in the periphery of the intracellular surface in the $G\alpha_s$, $G\alpha_i$ and $G\alpha_q$ coupled receptors. Several of the common contacts also contribute to G protein selectivity by differentially modulating the temporal frequency of the contact to couple members of different G protein subfamilies. We propose that for GPCR:G protein complexes, these common contacts must be satisfied for the interface of the complex to further stabilize and prolong the GPCR:G protein complex lifetime, aiding in G protein activation. Previously, differential evolution analysis of paralogous and orthologous G protein sequences identified a G protein-based barcode for selectivity[14,41]. Similar approaches have been applied to GPCRs, focusing on interfacial residues identified from three-dimensional structures of GPCR:G protein complexes. Identifying such a "selectivity" bar code based on consensus sequence has proven intractable, as GPCRs are more divergent in sequence than the G proteins[14,45]. This exemplifies the importance of both spatial and temporal properties of GPCR:G protein contacts playing an important role in G protein selectivity.

When performing experimental validation of the contacts identified from the spatio-temporal dynamics of GPCR:G protein complexes, we were concerned that single point mutations would have relatively small effects on coupling behavior within the large interface of the GPCR:G protein complex. Therefore, we focused on identifying G protein family specific contacts that were made by GPCR residues that were conserved among the two receptors within each G protein coupling class, but varied in the other receptors that couple to different G proteins (i.e. positions where residues are conserved in ADRB2 and ADORA2A, but vary in ADORA1, HTR1B, CHRM1, and/or HTR2A). Through this approach, we were able to identify positions and

substitutions in the Gq-coupled CHRM1 receptor that were able to decrease interaction with its primary cognate Gq protein and enhance interactions with secondary G proteins, Gs and Gi proteins.

In developing an LDA model of the GPCR:G protein residue contact space for Gs, Gi, and Gq, we believe that the availability of more simulation data from newly emerging GPCR:G protein complex structures can be continuously used to refine the boundaries between G protein classes. We observe that the D1DR:Gs data have stronger correlation to the Gs class sampled in the original LDA model and suspect this is because the Gs interaction may be highly similar across Gs-coupling receptors. The CB1R:Gi and H1HR:Gq show less correlation to their respective Gi and Gq classes from the original LDA model and in fact show strong correlation with Gq and Gi classes, respectively. In cryo-EM structures, we observe that the orientation of the G protein α5 helix are aligned in similar orientations for the Gi and Gq proteins complexed to their respective receptors. Because of these similarities, we suspect the conformational space of Gi and Gq coupled interactions may have similarities. Therefore, predictive models will benefit from refinements incorporating more structural models of each of these coupling interactions, especially using non-chimeric G protein sequences, as they become available.

Although promiscuous coupling of G proteins by GPCRs has been known it has been largely underappreciated. Our recent study predicted mutations in ADRB2 that enable it to couple and signal via $G\alpha_s$ and $G\alpha_q$ proteins with similar potency when stimulated by isoproterenol[33]. We identified that ADRB2 residue $Q229^{5\times68}$ forms a strong contact with $G\alpha_s$ residue $Q384^{H5.16}$. This contact pair is a common contact with particularly high frequency among Gs signaling complexes in our fingerprint dataset. We also observed a steric clash between the $G\alpha_q$ residue H5.22 (E355) and the ICL1/ICL2 regions of ADRB2. In that study, we mutated the ADRB2 5×68 position to tryptophan (Q229W) and the 34×54 position to lysine (Q142K) to mimic the interface chemistry found in the $G\alpha_q$ coupled Vasopressin 1 A Receptor. We showed this mutant became more amenable to coupling with the $G\alpha_q$ residues H5.16 (L349) and H5.22 (E355), maintaining this critical common contact of 5×68 with G.H5.16[33]. This provides evidence for using the spatiotemporal code to decipher structural determinants which mediate selective and promiscous GPCR coupling. As we showed in our previous work, coupling can be facilitated between GPCRs and non-cognate G proteins by removing certain selectivity filters and improving the interaction of residues in the common contact positions, facilitating complex formation between a GPCR and G protein pair. The promiscuous ADRB2-TM samples the G protein common contacts with higher frequency than both the Gs protein and Gq protein subfamily-specific contacts. We infer that GPCRs which evolved to be selective towards a G protein subfamily requires specific contacts to be made, thereby constraining the types of G protein to which they can couple to. GPCRs that evolved the ability to promiscuously couple likely reduce the number of constraining contacts that are found among selective GPCRs.

MD simulation is a key technique to provide insights into the role of the temporal sampling of GPCR:G protein contacts since the persistence of these contacts fall within the MD simulation time scale. One caveat to be noted is that the three-dimensional structures of the GPCR:G protein complexes form the initial models used in this study, and therefore the analysis here is limited to the resolved regions of these three-dimensional structures. However, intrinsically disordered intracellular loops are known to modulate G protein selectivity[46] and many of those determinants are not included in this study. In addition, the GPCR:G protein complexes could sample distinctly varied conformations[30] that are not considered in this study.

Based on the regions resolved in the individual G proteins, we observe that the C-terminal H5 helix is the main contributor to contacts formed between GPCR and $G\alpha$ protein. We also observe that the H5 helix residues represent most G protein residue positions that

contribute to selectivity. In future structural studies, we may identify new specificity determinants in difficult-to-resolve GPCR loop regions, particularly ICL3, with sites on the G protein outside of the H5 helix. This will improve efforts to model the dynamics of coupling selectivity. Still, in previous reporting by us and others, the H5 helix is shown to be sufficient for altering the selectivity of GPCRs to specific G proteins[7,8,14,33,41,47,48].

The spatiotemporal code derived in this study can be used to guide the design of mutants that stabilize interactions between GPCRs and different G proteins. Using the selective contacts identified for different G protein subfamilies in the spatiotemporal code, one can modify these contact positions in a promiscuous GPCR or G protein to improve affinity in a targeted manner to stabilize the GPCR:G protein interface. This can aid efforts to stabilize GPCR:G protein complexes for structure determination and pharmacological targeting as well as in interpreting the effects of natural variation and disease mutations at these positions within the human population.

## Methods

### Calculation of promiscuity index for the 267 GPCRs

The G protein coupling data from Inoue et al. and the Guide to Pharmacology database was obtained from GPCRdb. We used the categorical definitions of "primary" and "secondary" coupling for each G protein subfamily as provided within these datasets. The coupling data from Avet et al. "Table S1D: double normalized $E_{max}$ values" was used to define categorical G protein subfamily coupling scores by converting the $E_{max}$ values to "primary" (>0.8) or "secondary" ($0.2 < x < 0.8$) coupling. These qualitative scores were then converted into a numerical score consistently across all three datasets by assigning a value of "1" for "primary", "0.5" for "secondary", and "0" for non-couplers. For each receptor, we computed an average promiscuity across G protein subfamilies for each of the three datasets. We then applied a weight to these scores to distinguish how the coupling interactions were measured in the different datasets. Because the Avet et al. study measured coupling through BRET measurements, we weighted these interactions by multiplying each value by "4". The data obtained in the Guide to Pharmacology dataset is largely the result of manual curation of the literature, so we multiplied these values by a weight of "2". For interactions observed in the Inoue et al. dataset, in which the authors use a chimeric Gq protein to measure interactions, we applied a score of "1". A composite "promiscuity index" was derived for each GPCR by then taking the average of these weighted indices.

### Receptor model and ligand preparation

We have prepared all nine GPCR:G protein complexes for MD simulations starting from respective X-ray crystal or Cryo-EM structures (ADRB2 with Gs from pdb code of 3SN6[19], ADORA2A with mini-Gs from pdb code of 6GDG[20], ADORA1 with Gi2 from pdb code of 6D9H[26], HTR1B with Go1 from pdb code of 6G79[21], CHRM1 with G14 from pdb code of 6OIJ[27], HTR2A with chimeric mini-Gq from pdb code 6WHA[49], DRD1 with Gs from pdb code 7JVP[50], CNR1 with Gi from pdb code 6N4B[51], and HRH1 with Gq from pdb code 7DFL[52]. The mutations in ADRB2 (M96T, M98T, and N187E) were mutated back to the wild-type residues using Maestro (Schrödinger Release 2020-1: Maestro, Schrödinger, LLC, New York, NY, 2020.). Following mutations were done to convert G proteins to wild type: GNAS (7JVP) - A226G, S366A; GNAQ (7DFL) - T10C, A13E, D15A, A17E, V19R, E20R, R21I, S22N, K23D, M24E, D26E, N28Q, E31R, G33K, E34R, K35D, L324I. We added the missing sidechain residues and loops with fewer than five missing amino acids to the three-dimensional structures of the GPCR:G protein complex (Gαβγ heterotrimer included). The GPCR:G protein complex was embedded in explicit 1-Palmitoyl-2-oleoyl-sn-glycero-3-phosphocholine (POPC) bilayer membrane and solvated with water containing 0.15 M NaCl. Residues within 5 Å of the sites of mutation were minimized using MacroModel (Schrödinger Release 2020-1: MacroModel,

Schrödinger, LLC, New York, NY, 2020.) with position restraints on all backbone atoms. We have prepared all force field parameter for agonists (adenosine for ADORA1 and ADORA2A, BI-167107 for β2AR, donitriptan for HTR1B, iperoxo for CHRM1, 25CN-NBOH for HTR2A, SKF83959 for DRD1, MDMB-Fubinaca for CNR1, and histamine for HRH1.) using PRODRG server[53]. We calculated ESP (electrostatic potential) charge using Hartree-Fock method in the quantum mechanics software suite Jaguar[54]. We used the 6−31 G* basis set to calculate ESP charges.

### MD simulations for GPCR complexes

All MD simulations were performed using GROMACS2019 package[55] with CHARMM36m force field[56] with TIP3 water molecules. GPCR:G protein complexes were embedded into POPC (1-Palmitoyl-2-oleoyl-sn-glycero-3-phosphocholine) bilayer by CHARMM-GUI[57]. All crystal waters were retained, and 0.15 mM of sodium and chloride ions were added to neutralize each system. Each of GPCR complexes was minimized in energy using the steepest descent method in GROMACS. The SETTLE[58] and LINCS[59] algorithms were used, for the bond and angle for water and all other bonds, allowing 2 fs of time step. A cutoff distance of 12 Å for nonbond contacts was introduced, and the PME (particle mesh Ewald) method[60] was used for long-range vdW interactions.

Each solvated complex was first equilibrated by performing 200 ps of MD at 310 K using NVT ensemble. In this step, the atoms of the complex were restrained in their positions using a harmonic restraining force with a force constant of 1000 kJ/mol/nm$^2$. The water molecules and lipid bilayer were allowed to move to optimize their packing around the complex. As the next step, the complex was further equilibrated in the constant pressure and temperature (NPT) ensemble using gradually reduced harmonic position restraint from 5 to 1 kcal/mol/Å$^2$ applied to all heavy atoms of protein (receptor with heterotrimeric G protein). In the final NPT equilibration run, all positional restraints were released and run for 10 ns. The final snapshot of the equilibration run was starting structure of production simulations. We performed five replica runs with different initial velocities with each run up to 800 ns after NVT equilibration followed by stepwise NPT equilibration. A combined 1000 ns ensemble trajectory made from timesteps 600 ns ~ 800 ns of each velocity was used for the analysis contained in this manuscript.

### Parsing pairwise intermolecular contacts between GPCR and Gα protein

MD simulation trajectories were concatenated as 1μs ensembles. These trajectories were stored as xtc coordinate files and used to characterize the landscape of pairwise intermolecular residue contacts taking place between GPCR and Gα protein during the simulation. To characterize the pairwise contacts made, we used the "getcontacts" python script library (https://www.github.com/getcontacts). Contacts are defined as: salt-bridge, <4.0 Å cutoff between anion and cation atoms; hydrogen bond, <3.5 Å cutoff between hydrogen donor and acceptor atoms as well as <70 degree angle between donor and acceptor; van der Waals, <2 Å difference between two atoms; pi-stack contacts, <7.0 Å distance between aromatic centers of aromatic residues and <30 degree angle between normal vectors emanating from aromatic plane of each residue; cation-pi contacts, <6.0 Å distance between cation atom and centroid of aromatic rink and <60 degree angle between normal vector from aromatic plane to cation atom. For each GPCR:G protein complex simulation, the water, ion, and lipid molecules were stripped from the trajectory file used for contacts analysis. Within the command-line prompt to execute the script, the atom selection groups were set to match the chain identifiers and sequence range of amino acid residues in the GPCR ("--sele") and Gα protein ("--sele2"), adding a qualifier to only consider sidechain atoms for both sets of residues. This allowed us to map the pairwise,

sidechain-sidechain interactions which contribute to binding in a "sequence-specific" manner. We performed this analysis for all eight simulations of the GPCR:G protein complexes (six original PDBs from crystal and cryo-EM, two modeled complexes of promiscuous GPCRs).

The output of the contacts analysis was then converted into a binary fingerprint for each simulation (Supplementary Data 9), using in-house python scripts to perform the one-hot encoding (Anaconda Software Distribution. Computer software. Vers. 2-2.4.0. Anaconda, Nov. 2016. Web. https://anaconda.com) The first script ("relabel_resconts.py") converted each GPCR and G protein pdb number into a corresponding generic residue number based on the GPCRdb numbering[40] and the Common G protein numbering[41]. The second script (contRes_fingerPrints.py) populates a dictionary; the keys are each pairwise GPCR:G protein contact pair, and the values are an array of "0" matching the length of frames within the trajectory. When parsing the contact list frame-by-frame, any frame containing the contact pair results in a replacement of a "0" to a "1" at the index position corresponding to the frame number. This dictionary is converted into a data frame object[61] with pandas 1.4.3. The getcontacts scripts were used to compute the frequencies of each GPCR:G protein residue Figure contact across all eight simulations.

### Computing the linear discriminant model and derivation of the spatiotemporal code for selectivity

The resulting data frames were used to build a linear discriminant analysis (LDA) classifier using the Scikit learn 1.1.1 package[62] to identify the key pairwise contacts that distinguish binding across the different G protein subfamilies. Data frames from the original 6 PDB complexes were used to build the model using jupyter notebook 5.0[63]. ADRB2 and ADORA2A data frames were concatenated into the "Gs" class. The "Gi" class was comprised of ADORA1 and HTR1B, and data frames from the CHRM1 and HTR2A data frames represented the "Gq" class of residue contacts. The data was filtered to omit any pairwise contacts that were found in GPCR regions that were not resolved in all three classes, resulting in a total of 746 unique contact pairs. The LDA was fit using singular value decomposition (svd) and transformed into a two-dimensional space of the first and second linear discriminants of the model.

There is a total of 293,757 timepoints resulting from the MD simulations of the six GPCR:G protein complexes, with 764 contacts used as features of this dataset. As an initial test of the model, we performed a random 80:20 split on the dataset to train the LDA classifier using 80% of the data and tested the classification of the remaining 20% (Supplementary Figure 4a). This trained classifier explained 66.38% of the variance within component 1, and 33.62% of the variance within component 2. When testing the classification of the 20% holdout data, we achieved an accuracy of 99.64% of correct classification (Supplementary Fig. 4b).

The scaling of each contact for "Component 1" and "Component 2" of the linear discriminant are given in Supplementary Data 4. The weight vectors for each class (separated as Gs, Gi, and Gq in this model) and the mean contact frequency of each pairwise contact is given for each G protein subfamily (Supplementary Data 5, columns "cGx" and "wGx"). We combined the weights and mean frequencies of the contacts into a composite score, which allows us to use the spatial importance and the temporal persistence to identify the most critical contacts for selective interactions (Supplementary Data 5, columns "wGx"). We used the top-10 residue contacts based on composite score to use in our "spatiotemporal code" of contact pairs that distinguish each G protein subfamily from one another.

The covariance matrix of features for the 764 contacts is given (Supplementary Data 7) along with the two-dimensional plotting coordinates ("component 1" and "component 2") for each fingerprint of the dataset (Supplementary Data 4).

### Calculation of average frequency of contact per residue for all receptors

For the simulated GPCRs and their coupled G proteins, we calculated the average frequency (percentage of MD snapshots) per contact for those within the subset of common contacts and those made in the subset of subfamily specific contacts for the cognate G protein subfamily for each receptor. We averaged the contact frequency across all of the common and subfamily specific contacts sampled by a given GPCR:G protein complex and recorded this as the average frequency shown in Fig. 5c.

### Calculation of average gnomAD frequencies and activity of ADRB2 mutants from deep mutational scan

gnomAD missense variant frequencies were downloaded for six GPCRs (ADRB2, ADORA2A, ADORA1, HTR1B, CHRM1, HTR2A) directly from the gnomAD 3.1 web portal (https://gnomad.broadinstitute.org/). Missense variants at the GPCR positions identified in the relevant G protein selective LDA code were identified, and we calculated the mean frequency of variation across each of these positions for each receptor. We then calculated the average frequency of variation across all the remaining residue positions within each receptor to determine the relative abundance of mutations at the selective GPCR positions within the representation of the general population.

For the analysis of ADRB2 deep mutational scan data, we obtained the dataset of processed mutant activity data from the Jones et al. 2020 eLife[44] "Additional files" repository. The data were stratified into each of the tested concentrations and reported for each amino acid mutant at every ADRB2 residue position. We calculated the activity of the ADRB2 mutants stimulated at the $EC_{100}$ (625 nM) concentration of isoproterenol for the Gs-selective residue positions found within the Gs-selective LDA spatiotemporal code, and we calculated the global average of mutant activity at each residue position across the entire dataset. Results were plotted using ggplot2_3.3.6 in R.

### Calculation of average interaction energies of common and specific contacts for all receptors

The interaction energy (IE) between receptors and their respective G-proteins was calculated with the GROMACS "energy" program. Resulting IE was the total nonbond energy from short-range (within 12 Angstroms) Van der Waals and Coulombic forces. Data are represented as the summed average energies for pairwise contacts from the common and G protein-specific interactions identified for each GPCR:G protein pair, calculated from the terminal 100 ns of each replicated MD simulation. Data are presented as mean ± SD from five replicate MD trajectories.

### Experimental reagents

Dulbecco's Modified Eagle Media (DMEM), fetal bovine serum (FBS) (#1968431), and other cell culture additives were purchased from Gibco, Life Technologies (Carlsbad, CA). Linear polyethylenimine MW 25000 (PEI) (#23966-1) was purchased from Polyscience, Inc (Warrington, PA). Coelenterazine 400a was purchased from Nanolight Technology (#70217-82-2) (Pinetop, AZ). Anti-HA-peroxidase rat antibody (3F10) (#12013819001) was purchased from Roche (Manheim, Germany). BSA was purchased from Fisher BioReagents (Hampton, NH). Carbachol (#C4382), Poly-L-ornithine, SIGMAFAST OPD (#P9187-50SET) and 16% paraformaldehyde (PFA) was purchased from Sigma-Aldrich (St. Louise, ML). YM-254890 (#257-00631) was purchased from FUJIFILM Wako Chemicals U.S.A. DNA oligonucleotides were obtained from Integrated DNA Technologies (Coralville, IA).

### Plasmids and constructs

Biosensors for Gq[64], Gi3[64], EPAC[42] are described previously. An HA-tag (YPYDVPDYA) was inserted at the N-terminus of human M1 receptor (CHRM1) by PCR cloning. The CHRM1 mutants were

generated via a two-fragment PCR strategy. Briefly, for each mutant, mutation containing plasmids in half were generated by stepdown PCR using forward or reverse site-directed mutagenesis primers and ColE1 reverse or forward primers, respectively in two separated PCR reactions. The methylated template DNA in the PCR reaction were digested with DpnI and the PCR products were purified, and assembled together via Gibson recombination. All the mutant's coding DNA were verified by DNA sequencing (Genome Quebec, CES). The DNA sequences of all the primers were given in Supplementary Table 1.

## Cell culture and transfections
HEK293T cells were cultured in DMEM supplemented with 10% FBS, and 20 µg/ml gentamicin. Cells were grown at 37 °C in 5% $CO_2$ and 90% humidity. Cells were seeded at a density of $2.0 \times 10^4$ cells per well in a white 96-well flat bottom plate (for BRET) or clear 96-well flat bottom (for ELISA) and simultaneously transfected with receptor and sensor DNA using PEI transfection reagent. Briefly, 150 ng of hM1 DNA along with either 250 ng of Gq polycitronic sensor DNA, 25 ng of Gai3-RlucII/ 60 ng of GFP10Gγ2/60 ng of Gβ1 DNA(Gi3 sensor) or 25 ng of EPAC sensor DNA (adjusted total DNA amount to 1 µg by pcDNA) in 100 µl of PBS were mixed with 100 µl of PBS containing 3 µl of PEI. After 20 min incubation, the DNA/PEI complexes were dispensed into cells in 96-well plates (~15 µl/well). All assays were performed 48 h post-transfection. Gq Polycistronic, Gi3, and EPAC biosensors were used to assess Gq, Gi, and Gs activity respectively.

## BRET measurements
HEK293T cells expressing receptor and BRET sensors were incubated for 1 h with Tyrode's buffer (140 mM NaCl, 2.7 mM KCl, 1 mM $CaCl_2$, 12 mM $NaHCO_3$, 5.6 mM D-glucose, 0.5 mM $MgCl_2$, 0.37 mM $NaH_2PO_4$, 25 mM HEPES, pH 7.4). Cells were stimulated with serially diluted carbachol from $10^{-8}$M to $10^{-2}$M, and the BRET signal was recorded using the Biotek Synergy 2 plate reader with filter set 410/80 nm (donor) and 515/30 nm (acceptor). Cells transfected with EPAC biosensor were pretreated with 500 nM of YM-254890 compound for 30 min before carbachol stimulations. Cell-permeable substrate coelenterazine 400a (final concentration of 2.5 µM) was added 3 min prior to BRET measurements. BRET ratios were calculated by dividing the intensity of signal emitted by acceptor over the signal of light emitted by donor. The data was fitted to 12-point concentration response curves and analyzed for its activity. The experiments were performed as three biological replicates of the 12 single dosages, performed on different days.

## Cell surface expression via ELISA
WT and mutant receptors were transfected into HEK293T cells in polyornithine coated clear bottom 96-well plate. On the day of the experiment, the cells were washed once with PBS and fixed with 4% PFA in PBS for 15 min. The cells were then blocked with 1% BSA in PBS for 1 h and then incubated with anti-HA HRP (1:1000 in 1% BSA/PBS) for 1 h. The cells were washed four times with PBS and 100 µl of SIGMA-FAST OPD solution was added into each well. After 10 min, 25 µl of 3 M HCl was added to stop the reaction. The plate was then read at 492 nm in the Biotek Synergy 2 plate reader. Specific signals were obtained by subtracting the signal from mock (pcDNA) transfected cells. The experiments were performed as 12 technical replicates of three biological experiments performed on different days.

## Data analysis and statistics
Statistical analyses were performed using GraphPad Prism 6 software using Student's t-test. P values as well as significance were reported for $logEC_{50}$ and Emax % differentials. The curves presented represent the best fits and were generated using GraphPad Prism software.

## Reporting summary
Further information on research design is available in the Nature Research Reporting Summary linked to this article.

## Data availability
All data and analysis scripts used during the current study are included with the article (and its supplementary information files). The molecular dynamics simulations datasets generated for the current study are available in the GPCRMD.org [https://submission.gpcrmd.org/home/] repository under the following IDs: 1190, 1200, 1201, 1203, 1204, 1207, 1209, 1212, 1215, 1214, 1218. Initial structures for MD simulations used the following structures from the Protein Data Bank [https://rcsb.org/]: 3SN6, 6GDG, 6D9H, 6G79, 6OIJ, 6WHA, 7JVP, 6N4B, 7DFL. Missense variation data was downloaded from the gnomAD [https://gnomad.broadinstitute.org/] v. 3.1 webportal for the following GPCRs: ADRB2, ADORA2A, ADORA1, HTR1B, CHRM1, HTR2A.

## Code availability
The data analysis scripts used in this study used commercially available software available in Anaconda version 4.10.3 (Pandas 1.4.3, Scipy 1.8.1, Scikit-learn 1.1.1, Matplotlib 3.5.1, Seaborn 0.11.2, Jupyter-notebook 5.0, R 4.1.2, ggplot2_3.3.6). The scripts themselves are included as the Supplementary Data files.

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

## Acknowledgements

The authors acknowledge financial support by the NIH National Institute of General Medical Sciences (R01-GM117923, R01-GM097261, N.V.), the UK Medical Research Council (MC_U105185859, M.M.B.), the American Lebanese Syrian Associated Charities (ALSAC, M.S., M.M.B.), the Lundbeck Foundation (R313-2019-526, D.E.G.), the Novo Nordisk Foundation (NNF17OC003126, D.E.G.), and the Canadian Institutes of Health Research (PJT-162368 and PJT-173504, S.A.L.).

## Author contributions

M.S., N.V., and M.M.B. designed the research study; S.L., J.H.L., N.M., S.G., and E.M. generated data for the study; M.S. performed the analysis for the research study; A.C. and Y.N. performed the BRET experiments. S.A.L., A.C., and Y.N. analyzed the BRET results. M.S., N.V., M.M.B., and D.E.G. interpreted the results of the study; N.V. S.A.L. and M.M.B. supervised the study; M.S., N.V., A.C., S.A.L., M.M.B., and D.E.G. wrote and edited the manuscript.

## Competing interests

The authors declare no competing interests.
