## [Peer Review File · Nature Communications]

Dynamic Spatiotemporal Determinants Modulate GPCR:G protein Coupling Selectivity and PromiscuityREVIEWER COMMENTS

Reviewer #1 (Remarks to the Author):

Sandhu et al. study GPCR:G protein coupling selectivity using molecular dynamics simulation. For this, they monitored the interaction frequencies between receptors and G proteins for 12 complexes. Based on frequencies of pairwise residue contacts of selective/promiscuous receptors as well as the application of linear discriminant analysis (LDA), they propose a spatiotemporal code for G protein selectivity. First of all, I want to acknowledge the effort of the authors to address this relevant research question. I completely agree that a dynamic viewpoint is required to obtain a better understanding of G protein coupling selectivity. However, I have several major concerns about the methodology, data and derived conclusions that need to be addressed by the authors.

1. System setup: Authors write describe that the systems were curated by "...added the missing sidechain residues and loops with fewer than 5 missing amino acids to the three-dimensional structures of the GPCR:G protein complex (Gαβγ heterotrimer included)." This curation protocol is not sufficient for the mOR (PDB ID: 6DDF) - a structure in which helix 8 is missing. The authors would need to clarify why this receptor was simulated without H8.

2. Production run: authors run 5 x 200 ns to accumulate 1 us per complex. In order to ensure an appropriate sampling of the receptor:G protein interface, authors should run at least twice as much or up to 1 us per replicate.

3. Data analysis: authors state that 5 x 200 ns were concatenated as 1μs ensembles. Then, they run the "getcontacts" python script to obtain pairwise contacts. Data are plotted as contact heatmaps in Figure 3c and 5a. I noticed that the authors seem to filter the contacts further for promiscuous receptors in Figure S7 as indicated by the figure caption: "Contacts shown in the table are filtered for those that are sampled with at least 20% frequency within at least one simulation." I assume that this procedure applies also to Figures 3c and 5a - otherwise data are not comparable. The authors need to clarify this.

4. G protein selectivity and the persistent frequency order: Authors write: "Surprisingly, the persistent G protein common contacts also contribute to G protein selectivity through modulating their sampling frequency. For example, the GPCR: G protein contact pairs 6x32:H5.26 and 5x58:H5.13 are common to all G protein families (Figure 3c) with the persistent frequency order being $G_i > G_s > G_q$. Hence by modulating the relative persistence frequency this common contact becomes a G protein selectivity determinant."

In my opinion, the ranking ($G_i > G_s > G_q$) is not meaningful as there is no consistent set of GPCR:G protein complexes. For instance, authors rank two G_s complexes with either 0% or 90% contacts higher than one G_q complex with 40% contacts (numbers are approximated from Figure 3c). Such ranking seems rather random and does not describe a meaningful trend.

5. Involvement of ICL3 in G protein contacts: The ICL3 can establish relevant contacts with the G protein as stated by the authors. Unfortunately, in most simulated GPCR complexes the ICL3 is not entirely included. This makes the analysis of ICL3 contacts in this manuscript tricky and most likely misleading. The authors will agree that the number of contacts of the ICL3 significantly depends on its experimentally solved length in terms of the number of residues. Hence, comparison and ranking with respect to other receptor segments is not advisable as done in Figure 3.

6. Authors write "There is a diminished role for contacts formed by intracellular loop3 (ICL3) within promiscuous GPCRs. This corroborates observations made by other studies that ICL3 is involved in G protein selectivity 37,38"

The authors cite here two selected studies for muscarinic receptors showing that the ICL3 is relevant for G protein coupling selectivity. However, this cannot be generalized as the scenario is much more complex. In other studies, they show that the ICL2 is important for G protein selectivity but not the ICL3. This has been shown for the CB1R (<https://www.ncbi.nlm.nih.gov/pmc/articles/PMC3010585/>) – a receptor that is included in the set of simulations by the authors.

7. To understand the GPCR:G protein contacts sampled in promiscuous GPCRs, the authors performed MD simulations on two promiscuous receptors: (a) A2AR and (b) the engineered Q142K-R228I-Q229W triple mutant of β 2AR.

Authors state that there is a "diminished role for contacts formed by intracellular loop3 (ICL3) within promiscuous GPCRs". This statement can be misleading as not the whole ICL3 is simulated. See also point 5.

8. Spatiotemporal G protein selectivity code. Authors state: "...we identified that β 2AR residue Q2295x68 makes selective contact with Gas residue Q384H5.16. This contact pair is a strongly selective contact for Gs signaling in our spatiotemporal code (Fig. 4c). In that study, we mutated the β 2AR 5x68 position to Trp (Q229W) and showed this mutant gained promiscuity to couple with the G α q protein 36. This provides evidence for using the spatiotemporal code to engineer selective GPCRs from promiscuous GPCRs or vice versa."

I find this conclusion confusing. Based on Figure 5a, the engineered β 2AR:Gs complex establishes 100% contacts for 5.68x68_G.HG5.13. According to Figure 4c, this receptor should be Gi selective and not promiscuous. The discrepancy between the spatiotemporal selectivity code and the experimental outcome suggests that the current code is limited. Authors should carry out additional mutational experiments to test further the suitability but also limitations of the proposed code and based on the outcome refine their code.

9. Authors write "The average frequency of G protein specific versus common contacts made per residue in the promiscuous GPCR: G protein complexes showed that the promiscuous GPCRs show more frequent sampling of the common contacts rather than those identified as G protein family specific contacts (Figure 5b). This difference is much smaller for the β 2AR-TM with Gs. The discrepancy of the β 2AR-TM likely results from the mutant receptor, binding Gs protein very similarly to the wildtype receptor hence shares Gs-specific contacts."

I am not sure what the authors try to suggest here: (a) The 3D model of the β 2AR-TM with Gs is not sufficiently relaxed and needs more relaxation time, or (b) there are exceptions that do not clearly follow the rule, for instance, if a mutant complex is too similar to the WT complex. If authors refer to point (a), I propose to use longer equilibration times (1 us) to see if the complex relaxes. In the case of (b), the authors need to comment on this more extensively why the established rule is not applying.

10. As good scientific practice and for data transparency, simulation files including starting structures, simulation protocols, and trajectories should be made available on an open online resource, e.g. www.gpcrmd.org

Minor comments:

In Figure 2, it says 0.1 M but according to the method part, it should be 0.15M.

In most figures in the main manuscript and supplemental material, the greek letter beta is not correctly reproduced.

Flareplots are not visible in the supplemental material.

Reviewer #2 (Remarks to the Author):

In this study, Sandhu et al. proposes a new spatiotemporal code governing GPCR-G protein binding selectivity by performing MD simulations on several GPCR-Gp complexes. By analyzing the temporal persistence of GPCR-Gp contacts and their spatial distribution, they conclude that specificity-conferring contacts are located on the edges of the interface, while Gp common contacts are located at the center. Interestingly, they find that two promiscuous GPCRs preferentially sample common contacts. Extensive analysis of GPCR-G protein sequences and static structures have not yet revealed the molecular code for coupling selectivity. Therefore, focusing on dynamic interactions represents a novel approach and the findings could lead to a paradigm shift and significantly impact the field of GPCR structure/signaling function. However, there are a number of points that the authors need to address prior to publication.

Major points:

1. To define GPCR promiscuity, the authors refer to the study by Inoue et al who use chimeric G-proteins where only the 6 most C-terminal residues match the G-protein of interest. However, multiple lines of evidence (GPCR-Gp structures, functional studies including the authors' own findings) indicate that several key hotspot binding residues are present further Nterminal on helix H5 (i.e. from H5.19 to H5.13) that are not modified and sampled in the chimera. Could this restricted sampling affect the promiscuity assignment? How do the GPCR promiscuities reported by Inoue correlate with those derived from other studies (mentioned by the authors in Fig. S1) that do not consider chimeras but native G-proteins? For example, unlike the Inoue study that report A2AR as a promiscuous receptor, A2AR is assigned as Gs selective in the extensive study by Avet et al. Can authors comment on that?

2. The observation that promiscuous GPCRs sample more common contacts than selective receptors is interesting but based on only 2 receptors, a B2AR mutant and A2AR. Also, only A2AR gave very clear distinctive signatures between common and selective contacts. Considering that A2AR may actually not be that promiscuous, the authors should analyze other GPCRs that were unambiguously assigned as promiscuous in several experimental studies to fully support their findings.

3. An additional potential issue with the analysis of promiscuous GPCRs is the lack of starting experimental structures for the A2AR-Go / G14 and B2AR-Gq bound complexes used for this part of the study. How were these complexes modeled? Considering that the patterns of dynamic contacts identified through MD simulations will be largely dictated by the starting conformation of the GPCR-Gp complex, the accuracy of these models is critical.

4. Overall, considering the level of uncertainty concerning the level of promiscuity of A2AR and the accuracy of the structural models, experimental cross-validation of some key predicted selective and common contacts uncovered by the authors would considerably strengthen their findings and support the conclusions. A few point mutations would be sufficient. As of right now, only one selective site was validated in their previous study (Sandhu et al., PNAS 2019).

5. As mentioned by the authors in Fig. S3, GPCR-Gp pairs can sample multiple conformations (see NTS1R for example). Therefore, their analysis and conclusions may be restricted to a specific subset of conformations and contacts that receptor and G

proteins can adopt. The manuscript would benefit from a discussion of this potential limitation.

6. Most of the analysis is based on a binary definition of contact persistence that is derived from an apparently ad hoc interpretation of contact frequency (i.e. threshold of 20%). Why can't the authors derive a pseudo energy scale from the contact frequency to unify their analysis and provide a more quantitative assessment of binding contact strengths?

7. The simple topological organization of the GPCR-Gp binding site that emerges from the study is reminiscent of many protein-protein binding interfaces with core residues encoding stability (through hydrophobic interactions and hydrophobic effects) and peripheral residues encoding selectivity (through polar/charged attractive and repulsive interactions). Can the authors further analyze their findings and elaborate on the chemical nature of their identified contacts? Such analysis would provide more precise rules for future GPCR-Gp engineering efforts.

Minor point: a linear representation of the G protein sequence would be useful in Fig.3c with the residue numbers (H5.XX) mapped from the C terminus for non GPCR experts.

Reviewer #3 (Remarks to the Author):

The manuscript by Manbir Sandhu et al. presents the sorting of GPCRs based on selectivity (and promiscuity) based on previous published experimental work (Inoue et al). Additionally they perform extensive MD simulations for 12 GPCR:G protein complexes (1 μ s/complex) to derive a spatiotemporal code where residues that are involved in G protein selectivity and promiscuity are proposed (for Gs, Gi/o and Gq). The results show that residues at the periphery of the GPCR:G protein interface are important for selectivity and the ones at the core of the interface are common to all GPCR:G proteins. The persistence and sampling of both of these types of contacts are important to determine selectivity and promiscuity. The work is highly relevant and brings a dynamic perspective into GPCR:G protein selectivity.

Some issues are:

1. The authors briefly mention the other experimental work where GPCR:G protein selectivity and promiscuity are sorted based on BRET/FRET experiment (Avet et al and Olsen et al). Although the authors state that the list is slightly different there are major differences involving the receptors under study including the A2AR, where it is highly promiscuous in the present study and highly selective for Gs in the Avet et al. study. The authors should analyze similarities and discrepancies and discuss the possible reasons with the aim to reach a consensus GPCR:G protein selectivity landscape.

2. Following the previous point, the A2AR is simulated in the current manuscript and assumed highly promiscuous, I wonder what the validity of the that data is since the Avet study locates the A2AR as highly selective. The authors should defend the A2AR as a promiscuous GPCR (provide additional experimental data or point to other references where there is experimental information on the promiscuity of the A2AR). However, the Inoue study is based on Gq proteins with a replaced Gs H5 helix and, as shown in the current study, around 40-50% of the contacts are not in the H5 helix, could this be a reason for the discrepancy in results?

3. The H5 helix has been traditionally used to understand GPCR:G protein selectivity. Based on the current results the authors should discuss (a few more sentences) what is the predicted role of the G protein H5 helix vs other regions of the G protein involved in coupling. Are the later ones involved in specificity/promiscuity? what are the most relevant contacts outside H5? (there are some insights mentioned but a few more

sentences would be useful).

4. It would be interesting if the authors discussed whether new interactions appear from MD simulations that are not present in the cryo-EM/xtal structures and, if so, what is their abundance/relevance for the current study.

5. Overall the authors conclude that promiscuous GPCRs use more the common core interface residues while highly specific GPCRs use peripheric specific contacts. I wonder whether the authors could analyze their data to understand whether there are elements of negative selection, i.e. contacts at the periphery of the interface that do not allow certain types of G protein to bind to a GPCR. Or whether they have looked at them and not found any.

Typos:

- Line 20: promiscuous and not promiscuous
- Line 64. There is a double period.
- The names of the receptors in Figures 3c, 5b, 5c, S3, S4a, S7b, S7c seem to have errors (some sort of font problem, likely from the conversion to PDF?).
- Figure S4b and S4d only show the outer labelling and not show the connectivities (I assume another PDF conversion problem?).
- Figure S5 legend, line 69, extra parenthesis at the end of the line.
- Figure S6a and S6b, half of the molecule is hidden by the upper figure, leaving only visible the labels of the residues (more PDF conversion issues???)

Remarks from the authors to all three reviewers:

We thank the reviewers for their in-depth and thoughtful suggestions. We have attempted to constructively address all of the comments raised by individual reviewers, and where applicable, refer to changes in the manuscript by including them with our response. Our responses are found in “blue font” with indented paragraphs, while reviewer questions and comments are found in “black font” and numbered, unindented paragraphs. To avoid issues with reading Greek symbols and subscripts in different pdf readers, we have changed the names of each GPCR to the HUGO Gene Nomenclature Committee symbol. Thus, the receptors used in this study include: ADRB2 (β 2AR), ADORA2A ($A_{2A}R$), ADORA1 ($A_{1}R$), HTR1B (5-HT_{1B}R), CHRM1 (M_1 AchR), and HTR2A (5-HT_{2A}R).

During our revision of this work, the GROMOS forcefield was withdrawn by the developers of the GROMACS MD simulation workbench. Hence, we performed new MD simulations on 6 GPCR:G protein complexes using CHARMM-36 forcefield. We chose 2 complexes from each G α protein subfamily (Gs, Gi and Gq bound GPCRs). For each GPCR:G protein complex, we performed 5 velocity runs, with each MD simulation run lasting 800 nanoseconds to 1 microsecond. In total, we have ~5 microseconds of simulations for each system. We have now performed the same analysis that were communicated in the initial manuscript.

It is reassuring that the concepts that emerged from the initial manuscript remain unchanged. (i) The spatiotemporal frequencies of the GPCR:G protein contacts play a critical role in G protein coupling strength and selectivity. (ii) There are two types of GPCR:G protein residue contacts: those that are common among G protein subfamilies (Gi, Gs, and Gq) and those that are specific to each G protein subfamily. We observe that both types of contacts contribute to G protein coupling selectivity by a GPCR. (iii) We have derived a spatiotemporal code for G protein selectivity that identifies contacts that distinguish GPCR interactions with different subfamilies. (iv) Dynamics of GPCRs coupled to their G proteins show that they sample the common contacts at a higher average frequency than specific contacts. This phenomenon is most pronounced for promiscuous GPCRs, suggesting that promiscuous GPCRs that evolve from a more selective ancestral GPCR might lose their selectivity determinants during the evolutionary process. GPCRs which are selective to a specific G protein subfamily may evolve from a more promiscuous ancestor by gaining selectivity determinants that constrain the receptor to a smaller profile of G proteins with which to couple.

Experimental validation of our spatiotemporal codes: The period (wave length) of non-bonded interactions between GPCR:G protein is in the order of tens of nanoseconds. Experimental techniques to probe the persistence of these high-frequency interactions, especially for membrane proteins such as GPCRs are not tenable yet. That is the reason we used MD simulations to identify the persistent GPCR:G protein interactions that contribute to selectivity. In the revision, we have compared our findings to recently published experimental deep mutational scanning data for ADRB2 (doi: 10.7554/eLife.54895) and found that all the GPCR residue positions that we identified as selectivity determinants are more sensitive to mutations than neighboring residue positions. We also analyzed the gnomAD human genome variant database (compiled from over 100,000 individuals) and found that the GPCR residues that confer selectivity show less variation in the human population compared to other neighboring positions predicted not be involved in selectivity in all 6 GPCR:G protein complexes. These analyses support our conclusions for the functional importance of the selectivity determining residues.

REVIEWER COMMENTS

Reply to Reviewer #1 (Remarks to the Author):

Sandhu et al. study GPCR:G protein coupling selectivity using molecular dynamics simulation. For this, they monitored the interaction frequencies between receptors and G proteins for 12 complexes. Based on frequencies of pairwise residue contacts of selective/promiscuous receptors as well as the application of linear discriminant analysis (LDA), they propose a spatiotemporal code for G protein selectivity. First of all, I want to acknowledge the effort of the authors to address this relevant research question. I completely agree that a dynamic viewpoint

is required to obtain a better understanding of G protein coupling selectivity. However, I have several major concerns about the methodology, data and derived conclusions that need to be addressed by the authors.

We thank the reviewer for recognizing the significance of our study, and the need for dynamics to be incorporated into models of GPCR: G protein coupling selectivity.

1. System setup: Authors write describe that the systems were curated by "...added the missing sidechain residues and loops with fewer than 5 missing amino acids to the three-dimensional structures of the GPCR:G protein complex ($G\alpha\beta\gamma$ heterotrimer included)." This curation protocol is not sufficient for the mOR (PDB ID: 6DDF) - a structure in which helix 8 is missing. The authors would need to clarify why this receptor was simulated without H8.

We agree with the reviewer that modeling Helix 8 of the GPCR is a critical structural feature that influences the dynamics of the GPCR:G protein complex. Since we had to rerun all the MD simulations, we chose 6 GPCR:G protein complexes aiming to balance the number of system in each G protein subfamily and have thus removed mOR from our dataset. We focused the new simulations on modeling prototypical GPCRs of the aminergic and nucleotide binding families of receptors and structures containing major structural regions such as helix 8 in the original PDB structure.

2. Production run: authors run 5 x 200 ns to accumulate 1 us per complex. In order to ensure an appropriate sampling of the receptor:G protein interface, authors should run at least twice as much or up to 1 us per replicate.

We agree with the reviewer that sufficient simulation time is needed to ensure sampling of the GPCR:G protein interface and we have now run up to 800ns–1 μ s per replicate for all the systems reported here. This adds up to 5 μ s of aggregated trajectories for each system.

3. Data analysis: authors state that 5 x 200 ns were concatenated as 1 μ s ensembles. Then, they run the "getcontacts" python script to obtain pairwise contacts. Data are plotted as contact heatmaps in Figure 3c and 5a. I noticed that the authors seem to filter the contacts further for promiscuous receptors in Figure S7 as indicated by the figure caption: "Contacts shown in the table are filtered for those that are sampled with at least 20% frequency within at least one simulation." I assume that this procedure applies also to Figures 3c and 5a - otherwise data are not comparable. The authors need to clarify this.

We appreciate the reviewer for highlighting this potential discrepancy and allowing us to clarify the procedure we used. We have adjusted our figures with newly generated data and ensured consistency in analysis methods across all comparisons. The contacts labeled as "common contacts" or "subfamily specific contacts" are those that are sampled with at least 20% frequency within a representative system from each G protein subfamily (Gs, Gi, or Gq). The contacts used to build the Linear Discriminant Model have not been filtered by a frequency cutoff, because some of the low frequency contacts may contribute to selectivity and not to G protein coupling strength.

4. G protein selectivity and the persistent frequency order: Authors write: "Surprisingly, the persistent G protein common contacts also contribute to G protein selectivity through modulating their sampling frequency. For example, the GPCR: G protein contact pairs 6x32:H5.26 and 5x58:H5.13 are common to all G protein subfamilies (Figure 3c) with the persistent frequency order being $G_i > G_s > G_q$. Hence by modulating the relative persistence frequency this common contact becomes a G protein selectivity determinant."

In my opinion, the ranking ($G_i > G_s > G_q$) is not meaningful as there is no consistent set of GPCR:G protein complexes. For instance, authors rank two Gs complexes with either 0% or 90% contacts higher than one Gq

complex with 40% contacts (numbers are approximated from Figure 3c). Such ranking seems rather random and does not describe a meaningful trend.

We thank the reviewer for these valid comments and have removed these statements from the manuscript. We understand that the ranking is complex and, as the reviewer has mentioned, may not reflect similar frequencies for each GPCR:G protein complex within a specific G protein subfamily, hence we have removed these claims from the manuscript.

5. Involvement of ICL3 in G protein contacts: The ICL3 can establish relevant contacts with the G protein as stated by the authors. Unfortunately, in most simulated GPCR complexes the ICL3 is not entirely included. This makes the analysis of ICL3 contacts in this manuscript tricky and most likely misleading. The authors will agree that the number of contacts of the ICL3 significantly depends on its experimentally solved length in terms of the number of residues. Hence, comparison and ranking with respect to other receptor segments is not advisable as done in Figure 3.

We agree with the reviewer, and the aim of this work is not to discuss the effect of missing structural elements from 3D structures. We have modified our analysis of contacts by only focusing on the GPCR regions that are resolved in the complexes with each G protein subfamily, and by avoiding comparisons of regions with incomplete structural information. We have also stated those caveats in the fifth paragraph of the “Discussion” section:

“MD simulation is a pivotal technique to provide insights into the role of the temporal sampling of GPCR:G protein contacts since the persistence of these contacts fall within the MD simulation time scale. One caveat to be noted is that the three-dimensional structures of the GPCR:G protein complexes form the initial models used in this study, and therefore the analysis here is limited to the resolved regions of these three-dimensional structures. However, intrinsically disordered intracellular loops are known to modulate G protein selectivity and many of those determinants are not included in this study. Additionally, the GPCR:G protein complexes could sample distinctly varied conformations³⁰ that are not considered in this study.”

6. Authors write “There is a diminished role for contacts formed by intracellular loop3 (ICL3) within promiscuous GPCRs. This corroborates observations made by other studies that ICL3 is involved in G protein selectivity 37,38”. The authors cite here two selected studies for muscarinic receptors showing that the ICL3 is relevant for G protein coupling selectivity. However, this cannot be generalized as the scenario is much more complex. In other studies, they show that the ICL2 is important for G protein selectivity but not the ICL3. This has been shown for the CB1R (<https://www.ncbi.nlm.nih.gov/pmc/articles/PMC3010585/>) – a receptor that is included in the set of simulations by the authors.

The reviewer's point is valid, but since the manuscript has been revised extensively, with two GPCRs in each G protein coupling category, we have removed those statements and have instead focused on clear evidence that the GPCRs use a “common contact” subset across all G protein subfamilies to interact with G proteins. These common contacts are critical for coupling interactions with G proteins, and residues found at the periphery of the GPCR:G protein complex are involved in selectivity towards different G proteins. We observe that the peripheral contacts arise from many regions within the GPCR, as shown as spheres in the revised Figure 4d, with the residues forming common contacts shown as pink surface.

Legend: d) GPCR residues involved in G protein selectivity are displayed as colored spheres on the surface representations of a Gs (ADRB2, red spheres, top), Gi (ADORA1, green spheres, center), and Gq-coupled (HTR2A, blue spheres, bottom) GPCR. Residues which are found in “common contacts” across G protein subfamilies are shown as a magenta colored surface.

7. To understand the GPCR:G protein contacts sampled in promiscuous GPCRs, the authors performed MD simulations on two promiscuous receptors: (a) A2AR [ADORA2A] and (b) the engineered Q142K-R228I-Q229W triple mutant of β 2AR [ADRB2]. Authors state that there is a “diminished role for contacts formed by intracellular loop3 (ICL3) within promiscuous GPCRs”. This statement can be misleading as not the whole ICL3 is simulated. See also point 5.

In the revision, we have focused only on the triple mutant ADRB2 as our promiscuous GPCR representative and on the role of “common contacts” versus “subfamily specific contacts”, as opposed to any specific structural domain of the GPCR. We omitted ADORA2A as promiscuous receptor due to conflicting experimental results reported in the literature (Avet et al, doi: 10.1101/2020.04.20.052027 and Inoue et al, doi: 10.1016/j.cell.2019.04.044)

8. Spatiotemporal G protein selectivity code. Authors state: “...we identified that β 2AR residue Q229^{5x68} makes selective contact with G α s residue Q384^{H5.16}. This contact pair is a strongly selective contact for Gs signaling in our spatiotemporal code (Fig. 4c). In that study, we mutated the β 2AR [ADRB2] 5x68 position to Trp (Q229W)

and showed this mutant gained promiscuity to couple with the Gαq protein³⁶. This provides evidence for using the spatiotemporal code to engineer selective GPCRs from promiscuous GPCRs or vice versa.”

I find this conclusion confusing. Based on Figure 5a, the engineered β2AR:Gs [ADRB2:Gs] complex establishes 100% contacts for 5.68x68_G.HG5.13. According to Figure 4c, this receptor should be Gi selective and not promiscuous. The discrepancy between the spatiotemporal selectivity code and the experimental outcome suggests that the current code is limited. Authors should carry out additional mutational experiments to test further the suitability but also limitations of the proposed code and based on the outcome refine their code.

We apologize for the confusion and clarify that, based on our re-analysis of MD results, the aforementioned contact, 5x68 and G.H5.16, has been redefined as a common contact that contributes to GPCR:G protein interaction across multiple subfamily of G proteins, not just Gs. For these types of contacts, we show that the chemical identity of the amino acids involved is critical for effective GPCR:G protein interaction. In the ADRB2:Gs complex, the polar Gln229^{5x68} interacts with a similarly polar Gln384^{G.H5.16}. However, in the Gq protein, the residue at G.H5.16 is a hydrophobic Leu, which would not complement the polar Gln229^{5x68} in ADRB2. In the cited study, we had mutated the ADRB2 residue to a hydrophobic residue to improve the interaction with the hydrophobic Leu349^{G.H5.16} in Gq protein.

Regarding experimental cross-validation, we did not perform additional mutagenesis, but have used recently available experimental data to validate our findings, as detailed below.

To validate the amino acid positions in GPCRs as contributing to selectivity, we used the recently published ADRB2 experimental deep mutational scanning dataset provided in Jones et al (doi: 10.7554/eLife.54895) to measure the mutational tolerance at each of the ADRB2 residues found among Gs-selective positions identified in our Linear Discriminant model. We examined the average activity of each mutant in response to EC100 isoproterenol and observed that mutants for positions in the Gs-selective LDA contacts show responses that are well below the global average activity of all mutants across all ADRB2 positions (1.75 activity units); in fact, mutants in most positions (34x50, 34x51, 34x53, 34x54, 5x68, 7x55) showed below 1.0 activity units. This data is shown in Figure 4f.

Legend: f) The GPCR residues found within the Gs-coupled LDA-determined contacts are shown on the x-axis. The barplot displays the average “Normalized Activity” (\pm SEM) of each position when mutated to each of 19 alternate amino acids in the ADRB2 and stimulated with 625nM isoproterenol, as quantified in Jones et al. 2020 eLife. The dashed horizontal line represent the global average “Normalized Activity” of 1.75 units, calculated from the pooled activity of all variants across all conditions of isoproterenol stimulation.

We have similarly analyzed the amino acid positions in all 6 GPCRs that are identified as G protein subfamily selective from the Linear Discriminant model. For these analyses, we compared the average per-residue variant allele frequency of SNPs identified amongst the gnomAD 3.1 database at GPCR residues found among the selectivity positions and compared these to the the average per-residue variant allele frequency of SNPs at GPCR positions not found in selectivity positions. In each GPCR, we identified that the selectivity positions are less polymorphic in the human population than non-selective positions, suggesting protection from variation at these critical sites in the protein. This data is plotted in Figure 4e.

Legend: e) Residues identified in the LDA model that distinguish binding to different G protein subfamilies were assessed from a population genetics perspective to determine the level of natural variation at these selectivity-conferring positions compared to the background variation at GPCR residue positions. The bar-graph displays the mean variant allele frequency per residue of the collective GPCR residues among LDA-determined contacts for each receptor compared to the mean variant allele frequency per residue in the remaining residues of each receptor. The x-axis is plotted in reverse logarithmic order, with taller bars representing lower overall allele frequencies.

9. Authors write “The average frequency of G protein specific versus common contacts made per residue in the promiscuous GPCR: G protein complexes showed that the promiscuous GPCRs show more frequent sampling of the common contacts rather than those identified as G protein subfamily specific contacts (Figure 5b). This difference is much smaller for the β 2AR-TM [ADRB2-TM] with Gs. The discrepancy of the β 2AR-TM likely results from the mutant receptor, binding Gs protein very similarly to the wildtype receptor hence shares Gs-specific contacts.”

I am not sure what the authors try to suggest here: (a) The 3D model of the β 2AR-TM with Gs is not sufficiently relaxed and needs more relaxation time, or (b) there are exceptions that do not clearly follow the rule, for instance, if a mutant complex is too similar to the WT complex. If authors refer to point (a), I propose to use longer equilibration times (1 μ s) to see if the complex relaxes. In the case of (b), the authors need to comment on this more extensively why the established rule is not applying.

We apologize for the confusion: indeed, we meant (a). We have already extended the MD simulation time for all of the GPCR:G protein complexes analyzed, which revealed that all of the GPCR systems sample contacts from the subset of “common contacts” with greater frequency than “subfamily specific” contacts. The ADRB2-TM:Gs samples Gs subfamily specific contacts at a much lower frequency than those sampled in ADRB2-WT: Gs. Thus, we believe that the extended MD simulation time used in this newer analysis has sufficiently relaxed the ADRB2-TM model, as suggested by the reviewer.

10. As good scientific practice and for data transparency, simulation files including starting structures, simulation protocols, and trajectories should be made available on an open online resource, e.g. www.gpcrmd.org

We agree with the reviewer and have made available all analysis scripts, jupyter notebook for the LDA, and all the data in the Supplementary information. We are also sharing our MD simulations through the online repository at gpcrmd.org.

Minor comments:

In Figure 2, it says 0.1 M but according to the method part, it should be 0.15M.

Thank you. We have removed references to the methods except for within the methods section of the manuscript to maintain consistency.

In most figures in the main manuscript and supplemental material, the greek letter beta is not correctly reproduced.

We appreciate this feedback and have re-inserted the Greek symbol “ α ” throughout the manuscript in the same font when referring to G proteins. To avoid this issue for GPCR names, we have changed all reference to GPCRs by using their HUGO gene symbols.

Flareplots are not visible in the supplemental material.

We appreciate this comment and we have removed these now.

Reviewer #2 (Remarks to the Author):

In this study, Sandhu et al. proposes a new spatiotemporal code governing GPCR-G protein binding selectivity by performing MD simulations on several GPCR-Gp complexes. By analyzing the temporal persistence of GPCR-Gp contacts and their spatial distribution, they conclude that specificity-conferring contacts are located on the edges of the interface, while Gp common contacts are located at the center. Interestingly, they find that two promiscuous GPCRs preferentially sample common contacts. Extensive analysis of GPCR-G protein sequences and static structures have not yet revealed the molecular code for coupling selectivity. Therefore, focusing on dynamic interactions represents a novel approach and the findings could lead to a paradigm shift and significantly impact the field of GPCR structure/signaling function. However, there are a number of points that the authors need to address prior to publication.

We thank the reviewers for recognizing the significance of our work and for the in-depth and thoughtful suggestions.

Major points:

1. To define GPCR promiscuity, the authors refer to the study by Inoue et al who use chimeric G-proteins where only the 6 most C-terminal residues match the G-protein of interest. However, multiple lines of evidence (GPCR-Gp structures, functional studies including the authors’ own findings) indicate that several key hotspot binding residues are present further Nterminal on helix H5 (i.e. from H5.19 to H5.13) that are not modified and sampled in the chimera. Could this restricted sampling affect the promiscuity assignment? How do the GPCR promiscuities reported by Inoue correlate with those derived from other studies (mentioned by the authors in Fig. S1) that do not consider chimeras but native G-proteins? For example, unlike the Inoue study that report A2AR [ADORA2A] as a promiscuous receptor, A2AR is assigned as Gs selective in the extensive study by Avet et al. Can authors comment on that?

We agree with these comments, particularly we believe that in the Inoue study, the restricted sampling of just the terminal 6 amino acids of the G protein limits the complexity of what we already know about determinants of G protein selectivity arising from different regions of the G protein structure. In our own previous works we have shown that G protein selectivity determinants from the C-terminal H5-helix are found outside of the 6 most terminal residues. In our revised manuscript, we used the results from Inoue et al, Avet et al and the GPCR database to come up with an index for promiscuity. However, the reviewer is correct in pointing out discrepancies between the two studies, and we do not feel equipped to comment extensively on those differences. We have nevertheless decided to assign a higher weighting to the measured interactions from Avet et al, as these authors used wildtype GPCRs and G proteins, whereas the Inoue et al used chimeric G proteins.

2. The observation that promiscuous GPCRs sample more common contacts than selective receptors is interesting but based on only 2 receptors, a B2AR [ADBR2] mutant and A2AR. Also, only A2AR gave very clear distinctive signatures between common and selective contacts. Considering that A2AR may actually not be that promiscuous, the authors should analyze other GPCRs that were unambiguously assigned as promiscuous in several experimental studies to fully support their findings.

We note the reviewers concern in characterizing ADORA2A as a promiscuous GPCR. In the revised study we have omitted ADORA2A as promiscuous receptor due to conflicting experimental results reported in the literature (Avet et al, doi: 10.1101/2020.04.20.052027 and Inoue et al, doi: 10.1016/j.cell.2019.04.044), and focused on the triple mutant of ADRB2, ADRB2-TM, which had been shown to couple promiscuously with Gs and Gq in our previous study (Sandhu et al, doi: 10.1073/pnas.1820944116). Furthermore, after extending MD simulations to 800-1000ns length for individual simulation replicates, we identified that all GPCR:G protein pairs sample the common contacts with higher average frequency than G protein subfamily specific contacts. We have added these findings to the manuscript results and discussion sections.

3. An additional potential issue with the analysis of promiscuous GPCRs is the lack of starting experimental structures for the A2AR-Go / G14 and B2AR-Gq bound complexes used for this part of the study. How were these complexes modeled? Considering that the patterns of dynamic contacts identified through MD simulations will be largely dictated by the starting conformation of the GPCR-Gp complex, the accuracy of these models is critical.

As explained in response to point 2, we have removed the simulations with ADORA2A and used only the triple mutant of ADRB2-TM as our representative promiscuous GPCR in the revision. There are starting structures for ADRB2:Gs that were used to model the triple-mutant ADRB2-TM with Gs. We used the structure of muscarinic receptor, CHRM1 (also a biogenic amine receptor), in complex with G11 (a member of the Gq subfamily) to model a starting structure for ADRB2-TM:Gq (PDB ID: 6OIJ, doi: 10.1126/science.aaw5188). We previously modeled and validated the orientation of the 25 amino acid residues of the Gq C-terminal H5-Helix in another study (Sandhu et al, doi: 10.1073/pnas.1820944116), showing an ability to switch the selectivity of ADRB2 toward Gq binding.

4. Overall, considering the level of uncertainty concerning the level of promiscuity of A2AR [ADORA2A] and the accuracy of the structural models, experimental cross-validation of some key predicted selective and common contacts uncovered by the authors would considerably strengthen their findings and support the conclusions. A few point mutations would be sufficient. As of right now, only one selective site was validated in their previous study (Sandhu et al., PNAS 2019).

As explained in response to point 2, we have removed ADORA2A from our revision. Regarding experimental cross- validation, we did not perform additional mutagenesis, but have used recently available large-scale experimental mutagenesis data to validate our findings, as detailed below.

Jones et al. have recently published deep mutational scanning data on ADRB2 (doi: 10.7554/eLife.54895). The authors mutated each position of ADRB2 to all 19 alternate amino acids and measured signaling activity through the Gs pathway at basal condition (no stimulation) or with different doses of agonist isoproterenol. We examined the average activity of each mutant in response to EC100 isoproterenol and observed that mutants for positions in the Gs-selective LDA contacts show responses that are well below the global average activity of all mutants across all ADRB2 positions (1.75 activity units); in fact, mutants in most positions (34x50, 34x53, 34x54, 5x68, 7x55) showed below 1.0 activity units. This is shown in Figure 4f.

Legend: f) The GPCR residues found within the Gs-coupled LDA-determined contacts are shown on the x-axis. The barplot displays the average “Normalized Activity” (\pm SEM) of each position when mutated to each of 19 alternate amino acids in the ADRB2 and stimulated with 625nM isoproterenol, as quantified in Jones et al. 2020 eLife. The dashed horizontal line represent the global average “Normalized Activity” of 1.75 units, calculated from the pooled activity of all variants across all conditions of isoproterenol stimulation.

We have also calculated the mean variant allele frequency of all residue positions identified as missense variants in the gnomAD 3.1 human population database for the 6 GPCRs studied here. We compared the average number of variants per amino acid residue identified among the > 100,000 individuals in the human population at positions which are part of the LDA spatiotemporal code for the given receptor, and those positions not found in the code. Overwhelmingly we observe that GPCR residues within the LDA spatiotemporal code show variation than the rest of the population, suggesting that these positions are protected and likely under purifying pressure as shown in the new Figure 4e.

Legend: e) Residues identified in the LDA model that distinguish binding to different G protein subfamilies were assessed from a population genetics perspective to determine the level of natural variation at these selectivity-conferring positions compared to the background variation at GPCR residue positions. The bar-graph displays the mean variant allele frequency per residue of the collective GPCR residues among LDA-determined contacts for each receptor compared to the mean variant allele frequency per residue in the remaining residues of each receptor. The x-axis is plotted in reverse logarithmic order, with taller bars representing lower overall allele frequencies.

5. As mentioned by the authors in Fig. S3, GPCR-Gp pairs can sample multiple conformations (see NTS1R for example). Therefore, their analysis and conclusions may be restricted to a specific subset of conformations and contacts that receptor and G proteins can adopt. The manuscript would benefit from a discussion of this potential limitation.

We acknowledge the caveats from using cryo-EM or crystal structures of GPCR:G protein complexes and the existence of distinct GPCR:G protein conformations under cellular conditions. We have discussed these caveats in the revision and have included this fifth paragraph of the “Discussion” here:

“MD simulation is a pivotal technique to provide insights into the role of the temporal sampling of GPCR:G protein contacts since the persistence of these contacts fall within the MD simulation time scale. One caveat to be noted is that the three-dimensional structures of the GPCR:G protein complexes form the initial models used in this study, and therefore the analysis here is limited to the resolved regions of these three-dimensional structures. However, intrinsically disordered intracellular loops are known to modulate G protein selectivity and many of those determinants are not included in this study. Additionally, the GPCR:G protein complexes could sample distinctly varied conformations³⁰ that are not considered in this study.”

6. Most of the analysis is based on a binary definition of contact persistence that is derived from an apparently ad hoc interpretation of contact frequency (i.e. threshold of 20%). Why can't the authors derive a pseudo energy scale from the contact frequency to unify their analysis and provide a more quantitative assessment of binding contact strengths?

This is a good suggestion. We have now calculated the energy of GPCR:G protein interactions averaged over the MD trajectories for both common and G protein subfamily specific contacts for each system. These calculations show that the common and specific contacts have similar contribution to GPCR:G protein interaction energy strength for the six GPCR:G protein systems studied. However, for promiscuous GPCRs, the common contacts contribute significantly more to the enthalpy of G protein coupling compared to specific contacts. These results are shown in Supplemental Figure S4b.

Legend: **b)** The contacts considered “common” or subfamily “specific” were collectively used to calculate the non-bond interaction energy (coulombic and van der Waals forces) from MD simulations using the “gm_x_energy” utility of Gromacs. Energies are plotted as kJ/mol for the collection of contacts. The error bars indicate standard deviation calculated from the 5 individual replicate 200 ns trajectories used to generate the 1 μ s ensemble.

7. The simple topological organization of the GPCR-Gp binding site that emerges from the study is reminiscent of many protein-protein binding interfaces with core residues encoding stability (through hydrophobic interactions and hydrophobic effects) and peripheral residues encoding selectivity (through polar/charged attractive and repulsive interactions). Can the authors further analyze their findings and elaborate on the chemical nature of their identified contacts? Such analysis would provide more precise rules for future GPCR-Gp engineering efforts.

We appreciate this thoughtful comment and have included (Supplementary Figures S2D and S2E) the sequence weblogo for GPCR and G protein positions within the subset of “common contacts” at the core of the binding interaction, which revealed a relatively high similarity within the chemical subspace of these residues.

The aims of this study are to analyze how structurally conserved features of GPCRs and G proteins are used differentially to form a network of contacts that facilitate selective coupling to different G protein subfamilies. We intentionally analyzed these contacts in a manner that is agnostic of the specific chemical nature of interactions formed by different residue pairs, and instead focused on the identity of contacting residues based on their common G protein numbering or common GPCR numbering. We believe that regardless of the amino acid types found in contact pairs, it is the preservation of the contact made between those positions in the GPCR and G protein that is integral to the selective interactions. Since this will be different for different receptor-G protein complexes this will be non-trivial to generalize. Nevertheless, by providing the common numbering positions for these residues, we are providing a general guide for how selective and promiscuous coupling occurs between GPCRs and G proteins. We invite readers to investigate these pairs in their GPCR:G protein complex of interest based on our annotation of which positions are shown to contribute to selective and common interactions. Such an analysis of the chemical nature of contacts can be considered for a future work, but we believe this will be beyond the scope and aims of the current work.

Minor point: a linear representation of the G protein sequence would be useful in Fig.3c with the residue numbers (H5.XX) mapped from the C terminus for non GPCR experts.

We thank the reviewer for this comment, and we have added a FASTA alignment of the relevant G protein C-terminal H5 helix as Figure 3e.

```
e
      G.H5.  1      6      11     16     21     26
>gnas2_human  TENIRRVFNDCRDIIQRMHLRQYELL
>gnai2_human  TKNVQFVFDAVTDVVIKNNLKDCGLF
>gnao_human   TNNIQVVFDVTDIIIANLNRGCGLY
>gnaq_human   TENIRFVFFAAVKDTILQLNLKEYNLV
>gna11_human  TENIRFVFFAAVKDTILQLNLKEYNLV
```

Reply to Reviewer #3 (Remarks to the Author):

The manuscript by Manbir Sandhu et al. presents the sorting of GPCRs based on selectivity (and promiscuity) based on previous published experimental work (Inoue et al). Additionally they perform extensive MD simulations for 12 GPCR:G protein complexes (1us/complex) to derive a spatiotemporal code where residues that are involved in G protein selectivity and promiscuity are proposed (for Gs, Gi/o and Gq). The results show that residues at the periphery of the GPCR:G protein interface are important for selectivity and the ones at the core of the interface are common to all GPCR:G proteins. The persistence and sampling of both of these types of contacts are important to determine selectivity and promiscuity. The work is highly relevant and brings a dynamic perspective into GPCR:G protein selectivity.

We thank the reviewer for recognizing the significance of this study.

Some issues are:

1. The authors briefly mention the other experimental work where GPCR:G protein selectivity and promiscuity are sorted based on BRET/FRET experiment (Avet et al and Olsen et al). Although the authors state that the list is slightly different there are major differences involving the receptors under study including the A2AR [ADORA2A], where it is highly promiscuous in the present study and highly selective for Gs in the Avet et al. study. The authors should analyze similarities and discrepancies and discuss the possible reasons with the aim to reach a consensus GPCR:G protein selectivity landscape.

We thank the reviewer for this comment. We used the results from Inoue et al, Avet et al and the GPCR database to come up with an index for promiscuity. We did not include the results of Olsen et al (doi: 10.1038/s41589-020-0535-8) as their study focused on a small number of GPCRs and further complicated integrating the different data sources. However, the reviewer is correct in pointing out discrepancies between the two studies of Avet and Inoue, and we do not feel equipped to comment extensively on or discuss potential reasons for those differences as they require expertise in understanding the nuanced differences of detection limitations and signal amplification across the techniques. We have omitted analysis of ADORA2A as a highly promiscuous receptor from the revision due to conflicting experimental results reported in the literature (Avet et al, doi: 10.1101/2020.04.20.052027 and Inoue et al, doi: 10.1016/j.cell.2019.04.044)

2. Following the previous point, the A2AR is simulated in the current manuscript and assumed highly promiscuous, I wonder what the validity of the that data is since the Avet study locates the A2AR as highly selective. The authors should defend the A2AR as a promiscuous GPCR (provide additional experimental data or point to other references where there is experimental information on the promiscuity of the A2AR). However, the Inoue study is based of Gq proteins with a replaced Gs H5 helix and, as shown in the current study, around 40-50% of the contacts are not in the H5 helix, could this be a reason for the discrepancy in results?

We agree with the reviewer that there is a large amount of discrepancy regarding the level of promiscuity in ADORA2A, and as explained in response to point 1, we have removed this receptor from the revision. We have focused on the triple mutant of ADRB2, which had been shown to be promiscuously coupled to Gs and Gq in our previous study (doi: 10.1073/pnas.1820944116).

As mentioned in response to Comment 1 by Reviewer 2, “We agree with these comments, particularly that the chimeric regions used to measure G protein selectivity in the Inoue et al study are limited in their ability to truly probe G protein selectivity. In our own previous works we have shown that G protein selectivity determinants from the C-terminal H5-helix are found outside of the 6 most terminal residues.”

3. The H5 helix has been traditionally used to understand GPCR:G protein selectivity. Based on the current results the authors should discuss (a few more sentences) what is the predicted role of the G protein H5 helix vs other regions of the G protein involved in coupling. Are the later ones involved in specificity/promiscuity? what are the most relevant contacts outside H5? (there are some insights mentioned but a few more sentences would be useful).

We appreciate this comment. In our spatiotemporal code for selectivity, ~25% of the G protein residues are outside of the H5 helix. Still, the majority of contacts with GPCR occur through the H5 helix, largely due to how H5 helix inserts into the GPCR intracellular interface upon complex formation. Previous studies have reported on the contribution of contacts outside of the terminal end of the G protein C-terminus (last 10 residues) to selective interactions (<https://www.pnas.org/content/116/24/12054>), but even in those studies, selectivity is mainly imported by the residues of the H5 Helix. We have added these considerations to the 6th paragraph of the revised discussion:

“Based on the regions resolved in the individual G proteins, we observe that the C-terminal H5 helix is the main contributor to contacts formed between GPCR and G α protein. We also observe that the H5 helix residues represent most G protein residue positions that contribute to selectivity. In future structural studies we may identify new specificity determinants in difficult-to-resolve GPCR loop regions, particularly ICL3, with sites on the G protein outside of the H5 helix. This will improve efforts to model the dynamics of coupling selectivity. Still, in previous reporting by us and others, it has been shown that the H5 helix is sufficient for altering the selectivity of GPCRs to specific G proteins^{7,8,14,36,38,40,41}.”

4. It would be interesting if the authors discussed whether new interactions appear from MD simulations that are not present in the cryo-EM/xtal structures and, if so, what is their abundance/relevance for the current study.

This is a great suggestion. In the revision, we list the contacts from the spatiotemporal code that are present in the starting PDB file for each receptor (Supplemental Table S3). We find that each class (Gs, Gi, Gq) shows variation as to how many selective contacts are identified within the original PDB structures, as described below and in the revision (page 11, fourth paragraph in section titled “*Deriving the Spatiotemporal Code for G protein Selectivity by GPCRs*”):

“For the Gs-selective LDA contacts, 5/10 contacts are formed within the original PDBs of either ADRB2 (PDB 3SN6) or ADORA2A (PDB 6GDG). Those contacts are sampled at frequencies of 23.9-99.6% of the simulation duration, while the contacts newly formed within the simulation are sampled at frequencies of 9.8-93.4% of the simulation. For Gi-selective LDA contacts, 4/10 contacts are found within one or both original PDB structures for ADORA1 (PDB 6D9H) or HTR1B (6G79). Those contacts are sampled at frequencies of 29.5-65.4% of the simulation, while newly formed contacts are sampled at 17.1-96.1% of simulation duration. Among these Gi selective contacts, the residue 5x71 was missing from the HTR1B PDB file. For Gq-selective LDA contacts, 2/10 contacts were found within the PDB structures of CHRM1 (PDB 6OIJ) and HTR2A (PDB 6WHA). Those contacts are sampled at frequencies of 70.0% and 89.1% of the simulations, and the newly formed contacts are sampled at frequencies ranging from 25.9-89.2% of the simulation.”

5. Overall the authors conclude that promiscuous GPCRs use more the common core interface residues while highly specific GPCRs use peripheric specific contacts. I wonder whether the authors could analyze their data to understand whether there are elements of negative selection, i.e. contacts at the periphery of the interface that do not allow certain types of G protein to bind to a GPCR. Or whether they have looked at them and not found any.

This is a great question, and one that we had addressed in a previous study (doi: 10.1073/pnas.1820944116). In that study, we showed that the G protein residue G.H5.22 of the Gq protein is unable to bind the ADRB2 intracellular interface because of electrostatic clashing with the ICL1/ICL2 regions of ADRB2; we thus suggest that the periphery of ADRB2 acts to negate Gq binding through this mechanism of steric hindrance that destabilizes the complex despite any contacts formed through the “common contact” interface. The triple mutations in ADRB2-TM collectively reduced that steric hindrance filter and improved binding through the common contact 5x68 with G.H5.16, and by removing the selectivity filter at ADRB2 position 34x54. We have revised the text to make that point clearly, on page 17 in the fourth paragraph of the discussion:

“Although promiscuous coupling of G proteins by GPCRs has been known it has been largely underappreciated. Our recent study predicted mutations in ADRB2 that enable it to couple and signal via $G\alpha_s$ and $G\alpha_q$ proteins with similar potency when stimulated by isoproterenol³⁸. We identified that ADRB2 residue Q229^{5x68} forms a strong contact with $G\alpha_s$ residue Q384^{H5.16}. This contact pair is a common contact with particularly high frequency among Gs signaling complexes in our fingerprint dataset. We also observed a steric clash between the $G\alpha_q$ residue H5.22 (E355) and the ICL1/ICL2 regions of ADRB2. In that study, we mutated the ADRB2 5x68 position to tryptophan (Q229W) and the 34x54 position to lysine (Q142K) and showed this mutant became more amenable to coupling with the $G\alpha_q$ residues H5.16 (L349) and H5.22 (E355), maintaining this critical common contact of 5x68 with G.H5.16³⁸”

Typos:

- Line 20: promiscuous and not promiscuous
- Line 64. There is a double period.
- The names of the receptors in Figures 3c, 5b, 5c, S3, S4a, S7b, S7c seem to have errors (some sort of font problem, likely from the conversion to PDF?).
- Figure S4b and S4d only show the outer labelling and not show the connectivities (I assume another PDF conversion problem?).
- Figure S5 legend, line 69, extra parenthesis at the end of the line.
- Figure S6a and S6b, half of the molecule is hidden by the upper figure, leaving only visible the labels of the residues (more PDF conversion issues???)

We thank the reviewer for bringing these typing errors to our attention. We have fixed them and revised the manuscript extensively.

REVIEWER COMMENTS

Reviewer #1 (Remarks to the Author):

The authors have carried out a major revision of their manuscript. I would have liked to be more positive, but unfortunately, I cannot promote the paper in its current form:

1. Significantly reduced dataset: in the original work, authors used 12 GPCR:G protein complexes. The dataset was reduced to 6 complexes which represent only half of the original work. This significantly lowers the value of this work. In this context, I am very curious if the described spatiotemporal G protein selectivity code holds also true for the removed receptor complexes. I would strongly suggest that the authors include the missing complexes again which can serve as a test set to validate the spatiotemporal code.
2. Classification of "strongly selective contacts" and "common contacts": In response to my comment 8 about the spatiotemporal G protein selectivity code, reviewers write that the contact, 5x68 and G.H5.16, has been redefined to a "common contact" that contributes to GPCR:G protein interaction across multiple subfamily of G proteins, not just Gs when carrying out a re-analysis of MD results. I would like to understand the origin of this initial misclassification to obtain confidence about the new result. Furthermore, I feel that to sufficiently convince the community about the current hypothesis including the proposed classification for "strongly selective contacts" and "common contacts", the authors require a systematic experimental validation of their findings (see next point).
3. Experimental validation: The findings, although interesting, still lack sufficient experimental validation. To prove that the proposed spatiotemporal G protein selectivity code is truly valid, I propose that the authors experimentally mutate up to 2-3 key positions and confirm their impact on coupling selectivity across different receptors (i.e., A1R, A2AR, 5HT1BR, 5HT2AR, etc. in addition to the well-studied B2AR). This would serve as an excellent proof of concept and provide the necessary validation for the presented findings to be published in Nature Communications.

Reviewer #2 (Remarks to the Author):

The authors have correctly addressed my concerns in the revised manuscript.

Reviewer #3 (Remarks to the Author):

The authors have satisfied all concerns.

We thank the reviewers for their constructive suggestions and comments. Our reply to each query is italicized.

Reply to Reviewer #1 (Remarks to the Author):

1. Significantly reduced dataset: in the original work, authors used 12 GPCR:G protein complexes. The dataset was reduced to 6 complexes which represent only half of the original work. This significantly lowers the value of this work. In this context, I am very curious if the described spatiotemporal G protein selectivity code holds also true for the removed receptor complexes. I would strongly suggest that the authors include the missing complexes again which can serve as a test set to validate the spatiotemporal code.

In the original submission we had presented MD simulation results on 8 cognate GPCR:G protein complexes (2 Gs coupled, 5 Gi coupled and 1 Gq coupled receptor) along with 4 simulations of 2 promiscuous GPCRs coupled to 2 non-cognate G proteins.

In this second revised version of the manuscript we use 6 complexes to derive the spatiotemporal code. As per the reviewer's suggestion, we then tested the model that we derived on 3 more GPCR:G protein complexes that was not included while deriving the model. For this revision, we performed simulations on these 3 complexes. This totals to 9 complexes. Finally, we also simulated a promiscuous receptor bound to 2 non-cognate G protein. Thus, in the revised manuscript, we now have 10 complexes with simulations and validation.

*We have shown that the new simulation data on three GPCR:G protein complexes not included in the training set for the LDA model, when plotted in the linearly deconvolved space of the 6 cognate GPCR:G protein complexes (LD1 and LD2), shows general agreement with classification into Gs, Gi, and Gq classes. We observe that the D1DR-Gs data have stronger correlation to the Gs class from the original LDA model because the Gs interaction may be highly similar across Gs-coupling receptors. For the CB1R-Gi1 complex, which was used in the original submission, as our Gi-coupled representative, and for the Gq coupled H1HR-Gq there is less correlation to their respective Gi and Gq classes from the original LDA model, and we suspect this may be due to the more promiscuous nature of CB1R and HRH1 receptors, as demonstrated in the promiscuity index (**Figure 1**). Additionally, predictive models will benefit from refinements incorporating more structural models of each of these coupling interactions as more structures of GPCRs with Gq (and not chimeras of Gq) become available. Importantly, we have also performed experimental validation in collaboration with Dr. Stephane Laporte as described in our response to query 3.*

2. Classification of “strongly selective contacts” and “common contacts”: In response to my comment 8 about the spatiotemporal G protein selectivity code, reviewers write that the contact, 5x68 and G.H5.16, has been redefined to a “common contact” that contributes to GPCR:G protein interaction across multiple subfamily of G proteins, not just Gs when carrying out a re-analysis of MD results. I would like to understand the origin of this initial misclassification to obtain confidence about the new result. Furthermore, I feel that to sufficiently convince the community about the current hypothesis including the proposed classification for “strongly selective contacts” and “common contacts”, the authors require a systematic experimental validation of their findings (see next point).

In regards to the contact 5x68_G.H5.16, our initial simulations, performed using the GROMOS54a7 force field (recently retracted by the developers) resulted as an “exclusive contact” to Gs coupled GPCR. Upon using the CHARMM36 forcefield, we observe this contact is common to the Gs, Gi, and Gq coupled GPCRs. As to why these two forcefields differed in forming the contact in Gi and Gq coupled simulations, we can only speculate that the GROMOS54a7 forcefield was unable to form stable contacts between these residue positions in Gi and Gq proteins with their respective receptors because these positions are generally occupied by hydrophobic and uncharged residues, whereas the Gs coupled interactions are formed with polar and charged residues at these positions.

More importantly, in two of our previous publications, we have showed that this contact is a selectivity hotspot for interactions with Gs and Gq proteins to their cognate receptors, ADRB2 and AVPR1A, respectively. In “Semack et al, JBC 2016 (<https://doi.org/10.1074/jbc.M116.735720>)”, we demonstrate that the 5x68_G.H5.16 contact is critical for interaction of ADRB2 with Gs and AVPR1A with Gq; In order for ADRB2 to couple with non-cognate Gq proteins we experimentally mutated the G.H5.16 residue position in Gq to match the Gs residue, L349Q to promote coupling, and for AVPR1A to couple to Gs proteins we mutated the G.H5.16 position in Gs to match the Gq residue, Q384L, to promote coupling. In “Sandhu et al, PNAS 2019 (<https://doi.org/10.1073/pnas.1820944116>)”, we mutated the ADRB2 residue Q229W^{5x68} to match the hydrophobic interface found in Gq coupling receptors and to promote its interaction with the Gq protein.

In regards to the systematic experimental validation of our findings, we have addressed this in the response to comment #3, below.

3. Experimental validation: The findings, although interesting, still lack sufficient experimental validation. To prove that the proposed spatiotemporal G protein selectivity code is truly valid, I propose that the authors experimentally mutate up to 2-3 key positions and confirm their impact on coupling selectivity across different receptors (i.e., A1R, A2AR, 5HT1BR, 5HT2AR, etc. in addition to the well-studied B2AR). This would serve as an excellent proof of concept and provide the necessary validation for the presented findings to be published in Nature Communications.

*We thank the reviewer for this suggestion. Despite being a computational lab and being challenged to find experimental collaborators to test our predictions during this COVID time, we benefited from the help of Dr. Stephane Laporte (McGill University) and his expertise in GPCR signaling using biophysical approaches, to performed experimental validation of the selectivity code. We used the muscarinic acetyl choline receptor CHRM1, which is a Gq coupled receptor, and different sets of Gq, Gi and Gs BRET sensors to assess changes in G protein selectivity coupling of predicted mutations on this receptors. We have identified 2 loss-of-function mutations that disrupt the cognate Gq coupling, 2 gain-of-function mutations that promote Gi-coupling of CHRM1, and 2 gain-of-function mutations that promote Gs-coupling of CHRM1. We have now added these results to **Figure 4** and included additional authors from the Laporte’s lab in the publication. The detailed description of these experiments is included in the manuscript under the sub-section title “Experimental validation of residue positions in the spatio-temporal code that modulate selectivity to cognate and non-cognate G proteins in CHRM1 and the new **Figure 4** is shown below for reviewers’ convenience. We also have additional figures in the Supporting Information.*

Figure 4: Identifying G protein selectivity determinants using linear discriminant analysis of spatiotemporal heat map of GPCR:G protein contacts. a) We used one-hot encoding to binarize

all 764 persistent pairwise GPCR:G α protein contacts from each frame of the MD simulations into an interaction fingerprint. All fingerprints were used to train a Linear Discriminant classifier that was then used to identify features (intermolecular contact pairs) that distinguish Gs, Gi, and Gq interactions from one another. The top-10 features for each class were used to generate a “spatio-temporal” code of selectivity conferring residues in GPCRs. **b)** The intermolecular contacts identified in the six MD simulations of GPCR:G protein complexes were used to model the distinguishing contacts that separate Gs, Gi, and Gq type interactions. Linear Discriminant Analysis was used to identify pairwise contacts that contribute most highly to a signature that describes GPCR:G protein binding for each G protein family. The projection of each frame of the MD simulations are shown projected into the 2-dimensional deconvoluted space (Component 1, Component 2). **c)** The pairwise contacts which contribute highly to the interaction signature of each G protein family are given in the spatio-temporal code shown. The 2-dimensional table gives the GPCR (left) and G protein (top) residues that are found in the distinguishing pairwise contacts. The squares are colored based on which G protein interaction these contacts are found to distinguish. One contact (‘3x53:G.H5.19’) is shared among Gi and Gq type interactions, and is colored in “cyan.” **d)** GPCR residues involved in G protein selectivity are displayed as colored spheres on the surface representations of a Gs (ADRB2, red spheres, top), Gi (ADORA1, green spheres, center), and Gq-coupled (HTR2A, blue spheres, bottom) GPCR. Residues which are found in “common contacts” across G protein subfamilies are shown as a magenta colored surface. **e)** BRET-based Gq-activation sensor was used to measure CHRM1 (WT, P139K and S126A mutants) activation of Gq heterotrimers in the presence of carbachol. **f)** BRET-based Gi-activation sensor was used to measure CHRM1 (WT, E221K and A424K mutants) activation of Gi heterotrimers in the presence of carbachol. **g)** BRET-based EPAC sensor was used to measure CHRM1 (WT, P139K and S126A mutants) activation of Gs signaling and cAMP accumulation in the presence of carbachol and 500nM of Gq-protein inhibitor YM-254890.

Reviewer #2 (Remarks to the Author):

The authors have correctly addressed my concerns in the revised manuscript.
We thank the reviewer for their acceptance of our work.

Reviewer #3 (Remarks to the Author):

The authors have satisfied all concerns.
We thank the reviewer for accepting our work.

REVIEWERS' COMMENTS

Reviewer #1 (Remarks to the Author):

The authors have comprehensively addressed my remaining concerns.